# The Contribution of Antimicrobial Peptides to Immune Cell Function: A Review of Recent Advances

**DOI:** 10.3390/pharmaceutics15092278

**Published:** 2023-09-04

**Authors:** Hanxiao Li, Junhui Niu, Xiaoli Wang, Mingfu Niu, Chengshui Liao

**Affiliations:** 1Luoyang Key Laboratory of Live Carrier Biomaterial and Anmal Disease Prevention and Control, College of Animal Science and Technology, Henan University of Science and Technology, Luoyang 471023, China; 220320181453@stu.haust.edu.cn (H.L.); 210321251339@stu.haust.edu.cn (J.N.); 2College of Basic Medicine and Forensic Medicine, Henan University of Science and Technology, Luoyang 471023, China; yangtuo43139@haust.edu.cn; 3College of Food and Bioengineering, Henan University of Science and Technology, Luoyang 471023, China; mingfuniu@haust.edu.cn

**Keywords:** antimicrobial peptides, immune cell, anti-inflammatory, immunomodulatory, inflammatory mediators, signaling pathways

## Abstract

The development of novel antimicrobial agents to replace antibiotics has become urgent due to the emergence of multidrug-resistant microorganisms. Antimicrobial peptides (AMPs), widely distributed in all kingdoms of life, present strong antimicrobial activity against a variety of bacteria, fungi, parasites, and viruses. The potential of AMPs as new alternatives to antibiotics has gradually attracted considerable interest. In addition, AMPs exhibit strong anticancer potential as well as anti-inflammatory and immunomodulatory activity. Many studies have provided evidence that AMPs can recruit and activate immune cells, controlling inflammation. This review highlights the scientific literature focusing on evidence for the anti-inflammatory mechanisms of different AMPs in immune cells, including macrophages, monocytes, lymphocytes, mast cells, dendritic cells, neutrophils, and eosinophils. A variety of immunomodulatory characteristics, including the abilities to activate and differentiate immune cells, change the content and expression of inflammatory mediators, and regulate specific cellular functions and inflammation-related signaling pathways, are summarized and discussed in detail. This comprehensive review contributes to a better understanding of the role of AMPs in the regulation of the immune system and provides a reference for the use of AMPs as novel anti-inflammatory drugs for the treatment of various inflammatory diseases.

## 1. Introduction

Antimicrobial peptides (AMPs) are generally small molecular polypeptides of 12–50 amino acid residues with antibacterial activity [1]. To date, more than 3100 natural AMPs have been discovered [2]. AMPs can be extracted from animals, plants, and microorganisms and can also be chemically synthesized. AMPs are an important component of the innate immune system [3]. There are many classification methods for AMPs. AMPs can be divided based on their biosynthetic mechanism into ribosomal AMPs and nonribosomal AMPs. Nonribosomal AMPs are mainly derived from bacteria and fungi and have a narrow antibacterial spectrum. Most research in the field of AMPs focuses on ribosomal AMPs, which have a wide range of sources and diverse structures. AMPs can also be divided based on their secondary protein structure into peptides with α-helices, peptides with β-pleated sheets, peptides with both α-helices and β-pleated sheets, and linear peptides rich in certain amino acids. Furthermore, AMPs can be divided into five categories based on their natural sources: animal-derived AMPs, plant-derived AMPs, insect-derived AMPs, microbial-derived AMPs, and synthetic AMPs [4,5].

AMPs have antibacterial, antifungal, antiviral, antimicrobial, and antitumor functions and participate in immune regulation [1]. The antimicrobial mechanisms of AMPs can be represented by the barrel-stave model, the carpet-like model, the toroidal pore model, and the detergent-like model. Charge (cationic or anionic), size, primary sequence, conformation, structure, hydrophobicity, and amphipathicity may affect the antimicrobial activity and action mechanism of AMPs. AMPs can also be classified into membrane-acting and non-membrane-acting peptides. Membranes of pathogens and host cells differ considerably in the structure and composition are thought to be the basis of specificity of AMPs toward a target cell. AMPs with a net-positive charge typically accumulate on the membrane surface, interact via electrostatic interactions with phospholipids of negatively charged bacteria, and disrupt the cell membrane, followed by release of the cell contents [6]. The outer leaflet of eukaryotic cells enriched in zwitterionic phosphatidylcholine and sphingomyelin phospholipids may in part explain their lower susceptibility to the action of AMPs than bacteria [7]. Synthetic analogs of natural AMPs have been expressed in transgenic plants as protection from infection. Recombinant AMPs have also been used as therapeutics in aquaculture, as food additives for livestock, and as preservatives in food.

In addition, AMPs participate in the immune regulation and inflammatory processes [8]. AMPs regulate proinflammatory and anti-inflammatory responses mainly by modulating signaling pathways, directly or indirectly recruiting effector cells, including phagocytes, enhancing intracellular and extracellular bactericidal effects, and promoting macrophage differentiation and dendritic cell maturation. AMPs can also activate immune cells, promote immune cell proliferation, promote the secretion of immunoglobulins and cytokines, promote wound healing, regulate autophagy and apoptosis, and perform other functions in the host immune system [9]. In this study, evidence related to the mechanisms of different AMP-mediated regulatory functions in immune cells, including those of macrophages, monocytes, lymphocytes, mast cells, dendritic cells (DCs), neutrophils, and eosinophils, is reviewed.

## 2. Regulatory Effect of AMPs on Macrophages

### 2.1. Regulation of Macrophages by Mammalian-Derived AMPs

There are two major categories of mammalian AMPs: cathelicidins and defensins. Cathelicidins are composed of cyclic, extension-helical, α-helix, and β-sheet structures. Defensins generally contain 29–45 amino acids, and six cysteine (Cys) residues engage in intramolecular disulfide bonds [10]. The cathelicidin family of AMPs is a class of host defense peptides that are widely present in vertebrates, have high-efficiency and broad-spectrum antimicrobial activity, and have inhibitory effects on Gram-negative and Gram-positive bacteria, fungi, mycoplasma, and viruses [11]. For example, cathelicidins inhibited the release of hyaluronan-induced macrophage inflammatory protein-2 (MIP-2) from mouse bone-marrow-derived macrophages in a CD44-dependent manner, but do not inhibit MIP-2 production induced by a macrophage-activating lipopeptide. Cathelicidins can interfere with the membrane-binding events mediated by hyaluronan [12]. BSN-37 is a peptide composed of amino acid sequences at positions 2–38 of Bca5, a cathelicidin AMP extracted from bovine tissues. It is nontoxic to RAW264.7 cells and can promote expression of CD80, CD40, major histocompatibility complex I (MHC I), and MHC II in vitro and also increase secretion of the Th1-type cytokines interleukin-2 (IL-2) and interferon-γ and the Th2-type cytokines IL-10 and IL-4 (Figure 1) [13]. Porcine AMP PR-39 was the first member of the cathelicidin family of AMPs to be found in pigs. PR-39 comprises 1784 bp, with four exons, and encodes 39 amino acid residues. PR-39 modulates the activity of macrophages by inhibiting apoptosis and reducing caspase-3 activity [14]. In 2008, Wen et al. successfully constructed the expression vector pIRES2-GFP/PR39 and induced its expression in RAW264.7 cells. The ability of macrophages expressing PR39 to kill and clear intracellular bacteria was significantly enhanced [15]. CAP11 is a cathelicidin AMP extracted from guinea pig neutrophils that can significantly abolish necrotic cell death via inhibition of lipopolysaccharide (LPS) binding to target RAW264.7 cells, suppressing the release of high mobility group proteins. However, CAP11 does not affect LPS-induced apoptosis [16].

LL-37 is the only AMP in the human cathelicidin family. It is an amphiphilic helical peptide composed of 37 amino acid residues. In addition to its broad-spectrum bactericidal effect, it plays an important immune-mediating role in the process of apoptosis (Table 1) [17]. LL-37 mainly exists in the cytoplasm of macrophages and regulates production of inflammatory factors in a dose-dependent manner. Cell damage can ensue when the concentration of LL-37 exceeds optimal levels [18]. LL-37 upregulates expression of MCP-1 in macrophages and in the mouse lung, expression of IL-8 in human A549 epithelial cells, and expression of MCP-1 and IL-8 in whole human blood without stimulating the proinflammatory cytokine tumor necrosis factor α (TNF-α). LL-37 also inhibits production of MIP-1α and IL-12 [19], and upregulates the chemokine receptors CXCR-4, CCR2, and IL-8RB. LL-37 induces release of leukotriene B4 through activation of P2X7 receptors (P2X7R), extracellular regulated protein kinases 1/2 (ERK1/2), and mitogen-activated protein kinase p38 (MAPK p38), as well as through activation of cytosolic phospholipase A2 and 5-lipoxygenase in human monocyte-derived macrophages. The internalization of LL-37 promotes production of thromboxane A2 by upregulating expression of COX-2 [20]. However, LL-37 inhibits LPS/ATP-induced macrophage pyroptosis, IL-1β expression, and formation of inflammatory corpuscles by neutralizing LPS and inhibiting the response of P2X7R to ATP. In addition, LL-37 suppresses activation of caspase-1 and pyroptosis in peritoneal macrophages induced by cecal ligation and puncture and increases intracellular levels of IL-1β, IL-6, and TNF-α [21,22]. Pyroptosis is a type of programmed cell death in which cells continue to swell until they rupture and their contents are released, activating a strong inflammatory reaction. In 2011, Liu et al. found that LL-37 promotes proliferation and phagocytosis of RAW264.7 cells treated with LPS and upregulates transcription levels of CD80, CD86, IL-1β, TNF-α, and Toll-like receptor 4 (TLR4) [23]. There have been many recent studies on the global immunomodulatory effects of LL-37, and it has also been confirmed that LL-37 can regulate production of inflammatory factors and inhibit apoptosis, among other functions. However, its mechanism of action is still unclear, and further research must be carried out to provide new evidence for clinical application of AMPs.

Defensins were originally isolated from the leukocytes of rhesus monkeys and are found in almost all lifeforms. To date, 6 α-defensin peptides and 31 β-defensin peptides have been identified; β-defensin peptides are mainly expressed in birds and mammals [17]. Coupling human β-defensin 2 (hBD2) with spike protein enhances expression of the antiviral molecules IFN-β, IFN-γ, MxA, PKR, and RNaseL and the immune inducible factors IL-6, IL-1β, TNF-α, and NOD2 [24]. hBD3 profoundly suppresses Pg-LPS-induced production of TNF-α and IL-6 in RAW264.7 cells in a dose-dependent manner. Moreover, hBD3 attenuates phosphorylation of p38 and ERK1/2 in the MAPK pathway in vitro [25]. Porcine β-defensin 2 inhibits LPS-induced protein kinase B (Akt) and nuclear factor kappa-B (NF-κB) activation and production of inflammatory mediators IL-12, IL-6, IL-1β, TNF-α, and NO [26,27]. Furthermore, β-defensin DEFB123 inhibits the release of TNF-α and the LPS-induced activation of the MAPK signaling pathway in RAW264.7 cells [28]. hBD130 reduces LPS-induced generation of NO, IL-1β, IL-6, and TNF-α in mouse macrophages [29].

The tryptophan (Trp)-rich AMP indolicidin isolated from bovine neutrophils and its analogs significantly suppress production of NO and expression of inducible nitric oxide synthase (iNOS) mRNA in macrophages [30]. JH3 is an analog of AMP P3 isolated from the bovine erythrocyte hemoglobin α subunit. JH3 decreases release of lactate dehydrogenase and the phosphorylation level of p65; it inhibits release of IL-2, IL-6, and TNF-α by inhibiting the MAPK p38 pathway in human and mouse macrophages during infection by *Salmonella* Typhimurium CVCC541 [31]. JH3 significantly enhances antimicrobial activities in macrophages and alleviates the damage caused by microbial infection. JH-3 inhibits the release of cytochrome C in the cytoplasm and decreases the expression levels of TNF-αR2, caspase-9 and caspase-8 to further weaken caspase-3 activation and reduce apoptosis [32]. BMAP-18 is an analog of bovine myeloid BMAP-27. BMAP-18 and its analogs significantly inhibit NO production and TNF-α release in LPS-stimulated RAW264.7 cells [33]. Bactenecin 5, a proline-rich bovine AMP, activates THP-1 cells and synergistically triggers the upregulation of TNF-α in the presence of *Mycobacterium marinum* [34]. Bovine lactoferricin downregulates LPS-induced proinflammatory cytokines, TNF-α, IL-6, NO, and iNOS by targeting the LPS-activated NF-κB and MAPK signaling pathways [35]. It has been confirmed that bovine lactoferrin peptide has an anti-inflammatory effect on macrophages. In future studies, it will be important to determine whether bovine lactoferricin peptide has an anti-inflammatory effect in animal models.

Lactoferrin is the most active antiviral AMP in the transferrin family, with a high degree of conservation among species [17]. The human lactoferrin peptide hLF1-11 promotes monocyte differentiation into macrophages driven by GM-CSF, mainly manifested as IL-10 production [36]. Hepcidin (hepatic antimicrobial activity, HEPC) is a highly conserved AMP specifically synthesized and secreted by the livers in mammals. Both HEPC1 and HEPC2 promote iron storage and suppress iron release in Ana.1 and RAW264.7 cells in vitro and directly affect macrophages and promote iron retention [37].

### 2.2. Regulation of Macrophages by Amphibian-Derived AMPs

Amphibians can also produce AMPs. To date, more than 1400 AMPs have been found in amphibian skin secretions, and the skin of different species contains different active AMP molecules. Amphibian-derived AMPs can be divided according to their structural characteristics into linear AMPs with α-helical structures and cyclic AMPs containing intramolecular disulfide bonds [38]. Chensinin-1 and Temporin-1CEa are natural AMPs extracted from the skin secretions of *Rana chensinensis*. The modified peptides Chensinin-1b and W3R6 were obtained by replacing the amino acid residues of Chensinin-1. At concentrations below the IC_80_, Chensinin-1b and W3R6 reduce CD86 expression and release of TNF-α, IL-6, IL-1β, and NO in M1-type cells, increase secretion of IL-10 and transforming growth factor β (TGF-β) in M2-type cells, and inhibit LPS-induced phosphorylation of NF-κB and MAPK pathway proteins in mouse macrophages [39]. Temporin-1Cea, LK2(6), and LK2(6)A(L), which are peptides derived from Temporin-1Cea, clearly inhibit mouse macrophage foaming by inhibiting uptake of oxidized low-density lipoproteins by RAW264.7 cells, suppress NF-κB and MAPK inflammatory pathway signaling protein phosphorylation during oxidized low-density lipoprotein-induced mouse macrophage foaming, and reduce release of TNF-α and IL-6 [40]. PN-CATH1 and 2 were identified in tissues from the black-spotted frog *Pelophylax nigromaculata* and exhibit potent anti-inflammatory activity by effectively inhibiting LPS-induced production of the proinflammatory cytokines IL-6, IL-1β, and TNF-α in mouse macrophages [41]. Temporin-1Tl is a 13-residue AMP derived from frogs. Temporin-1Tl analogs demonstrate higher anti-inflammatory activity than Temporin-1Tl, as evidenced by greater inhibition of production of TNF-α and NO and mRNA expression of iNOS and TNF-α in LPS-stimulated RAW264.7 cells [42]. OL-CATH1 and 2 were identified in tissues from the odorous frog *Odorrana livida*. At low concentrations, OL-CATH2 significantly inhibits LPS-induced transcription and production of the proinflammatory cytokines TNF-α, IL-1β, and IL-6 in mouse peritoneal macrophages. In contrast, OL-CATH1 does not exhibit any detectable antimicrobial or anti-inflammatory activities [43]. HR-CATH is a member of the cathelicidin family of AMPs identified in tissues from frogs that induces macrophage chemotaxis and enhances the respiratory burst [44]. Some studies have found that the cathelicidin-family AMPs derived from tissues of certain frogs have strong anti-inflammatory functions, and there are many AMPs with strong antibacterial activity in the skin secretions of frogs. The novel 30-residue peptide Nv-CATH from the skin of the frog *Nanorana ventripunctata* belongs to the cathelicidin family. Nv-CATH suppresses excessive and harmful inflammatory responses by repressing production of NO, IL-6, TNF-α, and IL-1β. Nv-CATH also modulates macrophage trafficking to infection sites by stimulating CXCL1, CXCL2, and CCL2 production [45].

TK-CATH is an anionic AMP extracted from the skin of Guizhou salamander. TK-CATH suppresses expression of the inflammatory mediators TNF-α, IL-6, and MCP-1 in mouse macrophages by inhibiting the LPS-activated MAPK pathway [46]. *Limulus* anti-lipopolysaccharide factor-derived peptide CLP-19 significantly reduces the level of TNF-α released from LPS-treated RAW264.7 cells [47].

### 2.3. Regulation of Macrophages by Insect-Derived AMPs

Insect-derived AMPs have a wide range of antibacterial activities and no cytotoxicity. To date, there are 310 types of insect-derived AMPs in the AMP database. AMPs are divided into three major categories based on composition, molecular weight, and amino acid structure: amphoteric α-helical AMPs (polypeptides rich in proline or glycine residues and peptides with disulfide bonds), β-pleated sheet/hairpin structures, and amphiphilic α-helical structures [11]. AMPs secreted by housefly larvae enhance the ability of mouse peritoneal macrophages to phagocytize chicken red blood cells [111]. Mt6 and its dextroisomer D-Mt6, analogs of MAF-1, were isolated from the instar larvae of houseflies. These AMPs drastically decrease expression of IL-β and TNF-α in LPS-stimulated macrophages. This capability is likely due to dual abilities to directly bind LPS and to inhibit the LPS-activated MAPK signaling pathways in macrophages [48]. Although this study demonstrated the mechanism by which D-Mt6 reduces inflammatory cytokine expression levels, it has not yet been confirmed how this AMP induces cell lysis and death. Further research on this mechanism needs to be performed.

*Hermetia illucens* is a globally important nutritional resource insect. AMP HI-3 extracted from *Hermetia illucens* significantly enhances the phagocytic function of RAW264.7 cells in a dose-dependent manner. HI-3 significantly inhibits LPS-induced differentiation of RAW264.7 cells and reduces IL-6, TNF-α, and IL-1β mRNA expression and secretion levels as well as production of NO, but increases the expression level of IL-10. HI-3 also significantly increases superoxide dismutase activity and the total antioxidant capacity of RAW264.7 cells. HI-3 primarily achieves its anti-inflammatory effects by inhibiting the activation of p65 and the NF-κB pathway [49,50]. This study preliminarily demonstrated the potential anti-inflammatory effect of HI-3, but how HI-3 enters macrophages is still unclear, as is the specific mechanism by which HI-3 inhibits inflammatory reactions. The insect-derived AMP cecropin melittin hybrid (CEMA) blocks LPS-induced expression of IL-1β, IL-6, TNF-α, MIP-1α and MIP-1β in RAW264.7 cells. CEMA also induces expression of a set of 35 unique genes, including genes involved in cell adhesion and apoptosis [51].

RLYLRIGRR-NH2 (peptide A) and RLRLRIGRR-NH2 (peptide B), defensins derived from the beetle *Allomyrina dichotoma*, suppress LPS-induced TNF-α mRNA expression in mouse peritoneal macrophages. Peptide A supplementation of macrophages cultured with LPS results in a significant decrease in NO and TNF-α production. Furthermore, NF-κB activation is blocked by peptide A [52]. Defensin 1 from the red flour beetle *Tribolium castaneum* decreases production and release of IL-1β, IL-6, IL-8, IL-10, IL-12p70, IL-23, and TNF-α in blood-derived macrophages [53]. CopA5 is composed of five amino acids (LLCIA) from the CopA3 peptide (LLCIALRKK, D-form) originally isolated from the Korean dung beetle. CopA5 significantly inhibits LPS-induced NO production, TNF-α secretion, and the phagocytic activity of peritoneal exudate macrophages and upregulates phosphorylation of signal transducer and activator of transcription-1 (STAT1) [54].

Scorpions also produce AMPs, such as ToAP3 and ToAP4, which are extracted from the venom of the Brazilian scorpion *Tityus obscurus*. These AMPs reduce TNF-α and IL-1β levels in bone-marrow-derived macrophages and dendritic cells but increase IL-10 levels. Furthermore, the observed reduction in TNF-α secretion before LPS inflammatory stimuli is associated with peptide-TLR4 interaction. ToAP4 enhances MHCII expression in bone-marrow-derived dendritic cells, whereas ToAP3 reduces expression of costimulatory molecules such as CD80 and CD86 [55]. FA1 is a synthetic AMP derived from scorpion venom. FA1 activates the phagocytic activity of macrophages and enhances expression of IL-10, IL-12p70, and TNF-α [56].

The Bombyx mori AMP CM4 significantly Inhibits mRNA and protein expression of TNF-α, IL-6, and NO in LPS-stimulated RAW264.7 cells [57]. Cecropin DH, synthesized by deleting the hinge region (alanine-glycine-proline) of cecropin AMP B from Chinese oak silk moths, reduces LPS-induced production of NO and TNF-α in RAW264.7 cells [58]. Papiliocin is a 37-residue cecropin-like peptide isolated from the swallowtail butterfly *Papilio xuthus* and exerts anti-inflammatory activities by inhibiting NO production, secretion of TNF-α and MIP-2 and the innate defense response mechanisms triggered by TLR pathways that culminate in the nuclear translocation of NF-κB [59]. Periplanetasin-5 reduce TNF-α and IL-6 expression and NO production in RAW264.7 cells. Periplanetasin-5 controls inflammation by inhibiting phosphorylation of MAPKs and reducing degradation of IκB [60]. Ak-N’ and Ak-N’m were designed based on a template sequence from the transcriptome of the spider *Agelena koreana*. The peptides downregulate expression of the proinflammatory mediators TNF-α, IL-1β, and IL-6. Ak-N’m was also shown to interact with macrophage surface receptors containing TIR domains to inhibit MyD88-dependent and interferon-β-dependent TLR4 signaling in LPS-stimulated THP-1-derived macrophages [61]. In 2022, Zhang et al. constructed a truncated cecropin B peptide, KN-17, which causes RAW264.7 cells to transform from the M1 phenotype to the M2 phenotype by downregulating proinflammatory factors and upregulating anti-inflammatory factors [62]. KN-17 inhibits the NF-κB signaling pathway by reducing IκBα and p65 protein phosphorylation while promoting IκBα degradation and p65 nuclear translocation [62].

### 2.4. Regulation of Macrophages by Plant-Derived AMPs

To date, more than 300 plant-derived AMPs have been discovered [112], but only PN5 has been studied with respect to the effects of AMPs on the immune function of macrophages. PN5 is derived from *Pinus densiflora* (pine needle), and consists of 11 amino acids. PN5 suppresses the proinflammatory cytokines TNF-α and IL-6 through the NF-κB and MAPK signaling pathways [63].

### 2.5. Regulation of Macrophages by Microbial-Derived AMPs

Microbial-derived AMPs can be divided into bacterial AMPs, actinomycete AMPs, and fungal AMPs according to the type of microorganisms from which they originate [113]. *Bacillus subtilis* is widely distributed in soil and rotting organic matter. The antibacterial substances produced by *B. subtilis* are mainly divided into two categories: antimicrobial proteins and antimicrobial lipopeptides. Sublancin is a glycosylated AMP isolated from *B. subtilis* 168. Sublancin modulates innate immunity by inducing the production of IL-1β, IL-6, TNF-α, and NO, enhancing phagocytosis and the cell death-inducing activity of methicillin-resistant Staphylococcus aureus in both RAW264.7 cells and mouse peritoneal macrophages. Macrophage activation by sublancin was found to be partly dependent on TLR4 and the NF-κB and MAPK signaling pathways [64]. CSP32 is an AMP oligomer of bacitracin isolated from *Bacillus* spp. CSP32 stimulates phagocytosis while inducing the typical M1-polarized macrophage phenotype and enhances expression and production of proinflammatory mediators, such as NO, TNF-α, IL-1β, MCP-1, and PGE2. Furthermore, CSP32-stimulated inflammatory mediators are mainly induced by the MAPK and NF-κB signaling pathways during M1 macrophage polarization. CSP32 markedly increases the number of Ca^2+^-positive macrophages while upregulating phospholipase C and activating protein kinase Cε. BAPTA-AM, a Ca^2+^ chelator, significantly suppresses CSP32-mediated phagocytosis, inflammatory mediator production, and NF-κB activation [65]. It has been confirmed that the immune enhancement effect of CSP32 depends on calcium signaling pathways, but the type of calcium channel involved remains unclear. Whether CSP32 has an immune enhancement effect on human M1 and M2 cells requires further research. MS15 is derived from *Bacillus velezensis*, which is found in fermented foods. MS15 inhibits production of reactive oxygen species (ROS) and NO in RAW264.7 cells and increases the translation and transcription levels of catalase, glutathione peroxidase, and superoxide dismutase [66]. Macrolactin [67,68] and Surfactin [69] are also extracted from *B*. *subtilis*. Macrolaction can inhibits production and expression of iNOS, COX-2 and IL-6 in LPS-induced macrophages. The surfactin can impair the antigen delivery function of macrophages by inhibiting activation of NF-κB, p38, JNK, Akt signaling pathways, and expression of MHC-II and co-stimulatory molecules.

In addition to *Bacillus* spp., there are AMPs produced by other bacteria that regulate macrophage immune function. Ohmyungsamycin A (OMSA) is a cyclic peptide derived from a marine bacterium belonging to the genus *Streptomyces*. OMSA treatment amplifies *Mycobacteroides abscessus*-induced expression of the M1-related proinflammatory cytokines TNF-α, IL-1β, CCL5, and IL-12p40 and iNOS synthase, and significantly downregulates arginase-1 expression in murine macrophages [70]. BacSp222 is produced by a commensal bacterium *Staphylococcus pseudintermedius* strain 222. BacSp222 can effectively enhance the levels of NO, TNF-α, IFN-β, IL-1α, IL-10, IL-27 and MCP-1 in macrophages induced by interferon through activation of the NF-κB pathway [71].

Beauvericin (BEA) is a cyclic hexadepsipeptide produced by the fungus *Beauveria bassiana*. BEA dose-dependently blocks production of NO in LPS-treated RAW264.7 cells without inducing cell cytotoxicity and inhibits activation of NF-κB pathway [72]. The mycotoxin enniatin B (EnnB) is predominantly produced by *Fusarium genera*, and is often found in grain. EnnB causes the G0/G1-arrest, M2-like macrophage differentiation, apoptosis and necrosis. In addition, it is able to enhance LPS-induced IL-1β release in macrophages [73].

### 2.6. Regulation of Macrophages by Avian-Derived AMPs

Avian-derived AMPs are expressed in the bone marrow, bursa of Fabricius, liver, lung, and respiratory tract, as well as other tissues, and are the main defense in resistance to pathogenic microorganisms in birds [114]. Chicken-derived cathelicidin-2 (CATH-2) inhibits activation of TLR2 and TLR4 in macrophages by binding to extracellular lipoproteins and LPS, respectively. In addition, it inhibits the release of IL-6, IL-1β, IP-10, and RANTES after stimulation with *Escherichia coli* [74]. CATH-B1 is a chicken-derived cathelicidin that enhances bacterial phagocytosis by macrophages and downregulates avian pathogenic *E. coli* (APEC)-induced gene expression of IFN-β, IL-1β, IL-6, and IL-8 in primary macrophages. In addition, CATH-B1-preincubated macrophages show significantly higher gene expression of IL-10 after APEC challenge and reduce TLR4-dependent activation by APEC [75]. Chicken-derived defense peptide fowlicidin-1 (6–26) is chemotactic specifically toward neutrophils but not monocytes or lymphocytes. Fowlicidin-1 activates macrophages by inducing expression of inflammatory mediators, including IL-1β, CCL2, and CCL3, and enhances surface expression of MHC II and CD86 on RAW264.7 cells [76]. cNK-2 is a derivative of cNK-lysin, a cationic amphiphilic AMP derived from chicken. CNK-2 induces CCL4 and CCL5 expression in macrophages and inhibits LPS-induced inflammatory responses by suppressing IL-1β expression. The immunomodulatory activity of cNK-2 was involved in the MAPK signaling pathway, including modulation of p38, ERK1/2 and c-JNK, as well as the internalization of cNK-2 in cells [77].

Other birds can also produce AMPs that have beneficial effects on macrophages. Duck leukocyte AMPs significantly increase the E rosette formation rate, CD3^+^ level, immune organ index, macrophage phagocytosis index, and CD41, CD4^+^, and CD8^+^ levels in mice [115]. DCATH 12-4 and dCATH 12-5 are AMPs rich in Trp and lysine (Lys) that were synthetically designed based on duck-derived cathelicidins that significantly decrease the generation of NO, iNOS, and TNF-α in LPS-induced macrophages. These AMPs prevent LPS from binding to the carrier protein lipopolysaccharide-binding protein or to CD14 receptors of macrophages [78]. AvBD8, chicken avian β-defensin 8, stimulates expression of IL-1β, INF-γ, IL-12p40, CCL4, CXCL13, and CCL20 in macrophages and activates the MAPK signaling pathway through ERK1/2 and p38 signaling molecules [79]. It has been confirmed that poultry β-defensins play an important role in the host immune system, but their receptors have yet to be described. Further research may show the detailed mechanisms of action of poultry β-defensins.

### 2.7. Regulation of Macrophages by Other AMPs

AMPs are also an important component of the innate immune system of fish. Because fish rely more on innate immunity than adaptive immunity, fish AMPs can be used as a potential defense against emerging devastating infectious diseases [116]. To date, 49 AMPs have been isolated from fish [117]. Hepcidin is an AMP extracted from rainbow trout; it is taken up by RTS-11 cells and is able to induce expression of IL-10, IL-1β, and TNF-α at both transcriptional and protein levels [80]. Tilapia piscidin 4 (TP4), an AMP isolated from *Nile tilapia*, not only decreases levels of the proinflammatory mediators NO, TNF-α, IL-1β, and IL-6 in M1 macrophages but also enriches M2 macrophage markers. TP4 induces transformation of the M2 phenotype in M1 macrophages via the MAPK/ERK pathway and IL-10/STAT3 signaling [81]. Although the immunomodulatory activity of TP4 has been determined through in vitro models, in vivo studies are needed to verify the pharmacological potential of TP4. Epinecidin-1 (EPI) was extracted from orange-spotted grouper. EPI effectively suppresses LTA-induced production of the proinflammatory factors IL-6, COX-2, iNOS, and TNF-α in macrophages. Mechanistically, EPI attenuates LTA-induced inflammation by inhibiting TLR2 internalization and subsequent downstream signaling by ROS, Akt, p38, and NF-κB [82]. Another marine cathelicidin, Epi-1, inhibits bacterial toxins and LPS-induced proinflammatory cytokines in RAW264.7 cells. Epi-1 reduces expression of the proinflammatory factors TNF-α, IL-6, and IL-1β and increases that of the anti-inflammatory factors TGF-β and Sytx1. Moreover, Epi-1 induces expression of MHC-related genes, enhancing the immune response to LPS [83]. Scolopendrasin VII, derived from *Scolopendra subspinipes mutilans*, stimulates actin polymerization and subsequent chemotactic migration of macrophages through the activation of ERK and Akt activity [84]. Pardaxin is an AMP isolated from marine fish that induces THP-1 and U937 cells to differentiate into mature macrophages with phagocytic ability. It also increases expression of MyD88 in cells and reduces the phagocytic ability and superoxide anion production by leukemia cells [85]. LjPL-3 and LjPL-2, isolated from Japanese sea bass (*Lateolabrax japonicus*), reduce production of the inflammatory cytokines IL-1β, TNF-α, and TGF-β and promote macrophage chemotaxis and phagocytosis. LjPL-3 and LjPL-2 participate in the immune response by directly killing bacteria and activating macrophages [86]. An NK-Lysin peptide, NKHs27, derived from sevenband grouper (*Hyporthodus septemfasciatus*), enhances the respiratory burst of macrophages and upregulates immune-related gene expression [87].

With the development and advancement of genetic engineering technology, researchers have designed and optimized the coding sequences of AMPs with the assistance of biological information prediction and other related software and have obtained numerous of modified recombinant AMPs [118]. CNMs are novel nanocomposite antibacterial agents based on chitosan nanoparticles and the AMP microcin J25. CNMs significantly reduce the LPS-induced cytotoxicity of RAW264.7 cells. In one study, the LPS-induced inflammatory response was significantly ameliorated by CNMs through a decrease in levels of NO and proinflammatory cytokines, including TNF-α, IL-6, IL-8, IL-1β, TLR4, and the NF-κB and MAPK signaling pathways [88]. The hybrid peptides LF-KR and PaPMA significantly inhibited the generation and expression of NO and TNF-α in macrophages. In addition, PaPMA also inhibits expression of IL-6, IL-1β, and MIP-1/2 [89,90]. KR-1 and KR-2 are peptides optimized for 6K-F17. KR-1 and KR-2 restrains activation of the NF-κB signaling pathway by reducing the phosphorylation levels of IκBα and p65, inhibiting degradation of IκBα and nuclear translocation of p65, and increasing the proportion of the M2 phenotype among macrophages induced by LPS. KR-1 and KR-2 also inhibit production of iNOS, IL-1β, and TNF-α [91]. Furthermore, P18 is a novel AMP designed by hybridization of cecropin A and magainin 2; it inhibits the LPS-stimulated macrophage inflammatory response by directly binding to LPS and downregulating expression of iNOS, IL-1β, TNF-α, and NO and the production of NO [92]. The P18 analog is formed by replacing the N-terminal Trp 2 residues P18-W6, P18-W8, and P18-W15 and the center Pro9 with D-Pro or Nala (P18-Nala9, P18-D-Pro9) of P18. P18-W6, P18-W8, and P18-Nala9 significantly inhibits production of NO in RAW264.7 cells [93].

To date, many synthetic AMPs have demonstrated better antibacterial activity and stability than natural AMPs. GL13K is an AMP synthesized through artificial modification of a derivative of parotid gland-secreted protein. After coating a pure titanium surface with GL13K, a higher proportion of M1-type macrophages were found to be activated on the surface at the initial stage of culture, with M2-type macrophages being dominant at the later stage [94]. Moreover, expression levels of TNF-α and IL-1β were found to be decreased in M1-type macrophages, whereas levels of IL-10 and TGF-β3 were increased in M2-type macrophages [95]. KLW-f, a synthetic peptide rich in leucine and Lys, significantly inhibits NO production in RAW264.7 cells [96]. The AMP KLK and its analogs significantly inhibit LPS-induced production of NO, IL-1β, and TNF-α in RAW264.7 cells in a dose-dependent manner. KLK also suppresses production of PGE2, expression of iNOS and COX-2 mRNA, and mRNA and protein expression of IL-1β and TNF-α, as well as degradation and phosphorylation of IκB [97]. The synthetic peptide Kn2-7 enhances secretion of IL-10 and TNF-α in RAW264.7 cells, and the extent to which it enhances uptake of CpG DNA by cells is related to IL-10 secretion [98]. The synthetic peptide K_9_L_8_W, rich in Trp and Lys, inhibits production of NO, release of TNF-α, and expression of iNOS in RAW264.7 cells stimulated by low concentrations of LPS [99]. Lf6-pP is a symmetric-end AMP that reduces levels of TNF-α and IL-6 in LPS-stimulated macrophages [100]. The neutrophil membrane-coated nanoparticle-mediated KLA peptide KLA-NNPs inhibits macrophage phagocytosis in vitro and reduces the activity of intracellular bacteria and caspase-1 [101]. WK2 is a synthetic peptide with 10 short symmetrical β-hairpins that binds to endotoxin (LPS) and suppresses production of the LPS-induced proinflammatory cytokines iNOS, IL-8, IL-4, IL-1β, and TNF-α in RAW264.7 cells [102]. In 2023, Li et al. constructed a dual-functional polypeptide, LL37-C15. LL37-C15 inhibits release of the proinflammatory cytokines TNF-α, IL-1β, iNOS, and IL-6 by modulating the inflammatory pathway and reversing the M1 macrophage phenotype [103].

Cationic AMPs containing Trp inhibit the release of TNF-α and IL-6 in RAW264.7 cells induced by LPS without damaging cell morphology or the cell membrane [119]. CC34 and GW-A2 are both cationic peptides. CC34 significantly reduces production of TNF-α, IL-1β, and IL-6, levels of ROS, iNOS, and COX-2 mRNA, and phosphorylation of IKKβ, IκBα, and NF-κB in macrophages stimulated by LPS [104]. Many studies have shown that cationic polypeptides participate in multiple aspects of host immune defense. Future research should ultimately determine the mechanism by which cationic polypeptides inhibit excessive inflammatory responses based on their natural characteristics and activities in concert with the physiological environment in vivo. GW-A2 inhibits LPS-induced expression of NO, iNOS, COX-2, TNF-α, and IL-6 in macrophages, thereby reducing activation of the LPS-induced MAPK pathway and phosphorylation of protein kinase C-α/δ and increasing NF-κB activation [105]. The viability of tachyzoites treated with HPRP-A1/A2 was significantly decreased, with a reduction in adhesion to and invasion of macrophages by tachyzoites after HPRP-A1/A2 treatment. HPRP-A1/A2 damages the integrity of tachyzoite membranes, as characterized by membrane disorganization and cytoplasm outflow [106]. Cbf-14 is a cationic peptide derived from a cathelin-like domain that is highly conserved in the cathelicidin family. Cbf-14 decreases LPS-induced ROS secretion by blocking membrane translocation of p47-phox subunits and suppressing p47-phox protein phosphorylation. Additionally, the peptide downregulates overexpression of iNOS and finally inhibits excessive NO secretion from RAW264.7 cells stimulated by LPS. Moreover, Cbf-14 also downregulates the expression of p-IκB and p-p65 and inhibits nuclear translocation of NF-κB by blocking the MAPK and/or PI3K/Akt signaling pathways. Overall, Cbf-14 exhibits anti-inflammatory activity by inhibiting NF-κB activity and ROS production via the PI3K/Akt signaling pathway [107].

WALK11.3 is a model AMP with amphiphilic helicity. It has strong anti-inflammatory activity in LPS-stimulated RAW264.7 cells and inhibits expression of NO, COX-2, IL-1β, IL-6, INF-β, and TNF-α. Although WALK11.3 does not exert a major effect on all downstream signaling in the MyD88-dependent pathway, TLR4-mediated proinflammatory signals in the TRIF-dependent pathway were markedly attenuated due to inhibition of STAT1phosphorylation via attenuation of IRF3 phosphorylation. WALK11.3 specifically inhibits endocytosis of TLR4, which is essential for triggering TRIF-mediated signaling in macrophages [108]. In 2022, Kim et al. constructed two sets of WALK peptide isomers, WALK244.01-WALK244.10 and WALK243.01-WALK243.09. The peptides were shown to inhibit the TRIF-dependent signaling pathway in macrophages and production of iNOS, COX-2, and IL-1β [109]. Moreover, some AMPs used as feed additives significantly increase the phagocytosis rate and phagocytic index of RAW264.7 cells against chicken erythrocytes and enhance the nonspecific immune function of mice [120]. Keratin 6a-derived AMPs (KAMPs) suppress LTA- and LPS-induced NF-κB and IRF3 activation, proinflammatory cytokine production, and phagocyte recruitment, independent of their bactericidal function. Mechanistically, KAMPs not only compete with bacterial ligands for cell surface TLR and coreceptor (MD2, CD14, and TLR2) binding, but also reduce the cell surface availability of TLR2 and TLR4 by promoting receptor endocytosis [110].

## 3. Regulatory Effects on Monocytes by AMPs

### 3.1. Regulation of Monocytes by Mammalian-Derived AMPs

In recent years, nearly 100 types of AMPs have been found in mammals, including in mice, humans, pigs, horses, and cattle [121]. The cathelicidin peptide LL-37 is mainly synthesized in neutrophils and has an amphiphilic α-helix structure [122]. LL-37 combined with PGN induces monocytes to polarize toward the CD14^++^ and CD16^+^ monocyte subsets, ultimately leading to T cell proliferation and differentiation into Th17 cells (Figure 2) [123]. LL37-DNA complexes potentiate degranulation and IFN-γ production by natural killer (NK) cells upon subsequent encounter with K562 target cells. However, the complexes do not influence the NK cell proportion of peripheral blood mononuclear cells or expression of cytotoxic proteins by NK cells [124].

Defensins are a class of short cationic peptides rich in arginine residues and function in resisting invasion of various pathogenic microorganisms [125]. hBD1 promotes differentiation of monocytes derived from human umbilical cord blood into immature dendritic cells (DCs), including upregulating expression of immature CD40, CD80, CD86, and MHC II, downregulating expression of mannose receptors, and cooperating with LPS to promote the maturation of monocyte-derived DCs from human umbilical cord blood, including upregulating expression of CD83, CD40, CD80, CD86, and MHC II on the surface of mature DCs to promote proliferation of mature DC allogeneic lymphocytes [126]. hBD-3 induces monocyte activation through both P2X7-dependent (CD86 upregulation) and P2X7-independent (CD80 upregulation) signaling mechanisms, which raises the possibility that activation of P2X7R might play an important role in shaping the inflammatory microenvironment under conditions in which hBD-3 is highly expressed, such as in psoriasis or oral carcinoma [127]. hBD-3 induces the expression of the costimulatory molecules CD80, CD86, and CD40 on monocytes and myeloid DCs in a TLR-dependent manner. Activation of monocytes by hBD-3 is mediated by interaction with TLRs 1 and 2, resulting in signaling that requires MyD88 and IL-1 receptor-associated kinase-1 phosphorylation (Table 2) [128].

### 3.2. Regulation of Monocytes by Other AMPs

Liver-expressed AMP 2 (LEAP-2), an anionic AMP, plays a crucial role in the innate immunity of fish. LEAP-2 from *Boleophthalmus pectinirostris* promotes chemotaxis toward monocytes, enhances monocyte respiratory burst and bactericidal ability, inhibits overexpression of TNF-α and IL-1β, but has no impact on monocyte phagocytosis after treatment with *Edwardsiella tarda* [129]. However, current research on LEAP-2 expression is still in a preliminary experimental stage, and future research to explore LEAP-2 expression is warranted. Further research should also address the application of LEAP-2 in other aspects, such as aquaculture and feed additives. BpNKLP40 is an AMP extracted from *Boleophthalmus pectinirostris* that promotes mRNA expression of the proinflammatory cytokines TNF-α, IL-1β, and IFN-γ and inhibits that of the anti-inflammatory cytokines TGF-β and IL-10 [130].

CXCL14 is a CXC chemokine family member that contains an amphipathic cationic α-helical region at the C-terminus. The CXCL14-C17 analogs CXCL14-C17-a1 and CXCL14-C17-a3, which have improved cell selectivity, were engineered by introducing Lys, arginine or Trp into CXCL14-C17. CXCL14-C17-a2 and CXCL14-C17-a3 effectively inhibits production and expression of NO, TNF-α, IL-6, and MCP-1 in LPS-stimulated RAW264.7 cells [131].

## 4. Regulatory Effects on Lymphocytes by AMPs

### 4.1. Regulation of Lymphocytes by Mammalian-Derived AMPs

LL-37 can enhance delivery of CpG to B cells and pDCs but not T cells; this ability is independent of its amphipathic structure and its bactericidal property [132]. Furthermore, LL-37 induces apoptosis in CTLs and Treg cells through various mechanisms caused by leakage of lysozymes induced by LL-37 [133,134]. In one study, proliferation of CD4^+^ T cells was promoted for a certain period following cocultivation with hBD2 and hBD3 [135]. hBD3 alone inhibits IFN-γ-induced STAT1 tyrosine phosphorylation, but does not inhibit STAT1 serine and ERK1/2 threonine phosphorylation. hBD-3 activates T cells through these two signaling pathways, enhances T cell function and upregulates IL-2 and IL-10 production (Figure 3) [136].

Recombinant *Pichia pastoris* coexpressing porcine IL-4/6 and the porcine AMP fusion gene VRP, as well as recombinant *Pichia pastoris* with bovine AMP FBC, promotes the proliferation of porcine lymphocytes and increases the number of CD4^+^ T cells and CD8^+^ T cells in peripheral blood [137,138]. In addition, bovine leukocyte AMPs also increase the proportion of CD3^+^, CD4^+^, and CD8^+^ T lymphocytes in mouse peripheral blood [139]. BSN-37 enhances proliferation and activation ability of mouse spleen lymphocytes and significantly increases the production of IL-2, IL-4, and IL-10 (Table 3) [13].

### 4.2. Regulation of Lymphocytes by Amphibian-Derived AMPs

Frenatin 2.1S was first isolated from skin secretions of the frog *Sphaenorhynchus lacteus*. Frenatin 2.1S enhances the activation state and homing capacity of Th1-type lymphocytes and NKT cells in the mouse peritoneal cavity, as evaluated by increased expression of the early activation marker CD69 among T- and NKT cells and the chemokine receptor CXCR3 among T cells. Frenatin 2.1S significantly increases proportion of (F4/80^+^CD11c^+^CD206^+^) proinflammatory M1 macrophages and expression of MHC II molecules on F4/80^+^CD11c^+^ macrophages in the mouse peritoneal cavity [140].

### 4.3. Regulation of Lymphocytes by Insect-Derived AMPs

Cecropin and synbiotics significantly increase the number of lymphocytes in the jejunal epithelium; cecropin also increases antibody levels against Newcastle disease virus and increased the T lymphocyte transformation rate [141,142,143]. Synbiotics are biological preparations that combine one or several probiotics and prebiotics and are capable of simultaneously exerting the effects of both [148]. Moreover, the immunopeptide from silkworm pupa protein promotes proliferation of mouse B and T cells by inducing NF-κB activation. Conversely, it promotes balanced transformation of mouse Th cells to the Th1 type by increasing the production of IL-12, thereby exerting immunoregulatory effects [144].

### 4.4. Regulation of Lymphocytes by Microbial-Derived AMPs

Sublancin inhibits production of proinflammatory factors in *Salmonella*-infected mice, promotes production of IL-10, and enhances innate immune function. Furthermore, it significantly increases the ratio of CD4^+^/CD8^+^ T cells in mouse splenocytes to enhance cellular immune function [145].

### 4.5. Regulation of Lymphocytes by Other AMPs

The compound AMP “Taikanglibao” has been shown to improve the spleen index of weaned piglets, as well as reduce the number of splenic lymphocytes at the G_0_/G_1_ stage and the proportion of splenic lymphocyte apoptosis. In addition, it increased the number of splenic lymphocytes and the PI value of lymphocytes at the S stage and G_2_+ M stage, elevated the proportions of CD3^+^, CD3^+^CD4^+^, CD3^+^CD8^+^, and CD4^+^/CD8^+^ peripheral-blood lymphocyte subsets and the lymphocyte transformation rate, and promoted lymphocyte proliferation, improving overall cellular immune function [146]. The AMP LF-6 also improves the ability to resist bacterial infection by increasing the activity of mouse lymphocytes and inhibiting expression of the proinflammatory factors IL-1, IL-2, and IL-6 [147]. In 2023, Liu et al. found that the addition of AMPs to the diet improved the transformation rate of T cells [149].

AMPs extracted from the skin of the African ostrich increase the number of T cells in immune organs [150]. In 2007, Wang et al. reported that the crude extract of α-helix AMPs promoted the proliferation and transformation of peripheral blood lymphocytes in broilers [151].

## 5. Regulatory Effects of AMPs on Mast Cells

### 5.1. Regulation of Mast Cells by Mammalian-Derived AMPs

LL-37 is a chemotactic agent for mast cells that directly stimulates the degranulation of mast cells and production of the proinflammatory factors IL-4, IL-5, and IL-1β, while increasing that of TLR4 (Figure 4). In addition, LL-37-activated signal transduction in mast cells participates in the PLC/A2, MAPKs, and PI3K pathways (Table 4) [152,153]. LL-37 may bind negatively charged cell-surface molecules, become rapidly internalized into cells via clathrin-mediated endocytosis, and interact with MrgX2 to activate mast cells (LAD2 cells) [154]. Moreover, LL-37 activates MrgX2-induced degranulation of human mast cells and release of de novo-synthesized mediators. LL-37-induced LAD2 cells participate in degranulation and the release of IL-8, which is related to the activation of the G_i_ protein PLC/PKC/Calcium/NFAT, PI3K/Akt, and MAPK signaling pathways [155]. Defensin also induces release of IL-2, IL-4, IL-6, GM-CSF, nerve growth factor, PGE2, and LTC4 in mast cells and increases the expression of substance P mRNA [156]. Substance P is a neuropeptide widely distributed in fine nerve fibers. hBD2 stimulates mast cells through a G-protein phospholipase C-dependent mode, mobilizing intracellular Ca^2+^ and releasing histamine or PGD2 [157]. hBD-3 and hBD-4 activate mast cells by inducing MAPK p38 and EPK1/2 phosphorylation, increasing intracellular Ca^2+^ concentrations with degranulation and PGD2 release. Moreover. hBD-3, hBD-4 and LL-37 enhance the vascular permeability of mast cells [158].

### 5.2. Regulation of Mast Cells by Amphibian-Derived AMPs

Brevinin-2KP, an amphibian-derived AMP, is extracted from frog skin and promotes degranulation of mast cells and histamine release [159].

### 5.3. Regulation of Mast Cells by Insect-Derived AMPs

Api88 is a synthetically designed peptide based on the sequence of native apidaecin 1b, which inhibits the release of TNF-α and production of reactive oxygen intermediates in mast cells induced by LPS. In human mast cells, Api88 triggers degranulation and intracellular Ca^2+^ mobilization [160].

### 5.4. Regulation of Mast Cells by Other AMPs

AMP-IBP5 is a product of the cleavage of IGFBP-5 by serine proteases, which increases the content of Ca^2+^ in mast cells and plays an important role in mast cell degranulation and migration. AMP-IBP5 induces degranulation of mouse peritoneal mast cells via Mas-related G protein-coupled receptor-B2 (MrgprB2), a mouse counterpart of MrgprX2 [161]. Murepavadin is a lipidic HDP analog that induces mobilization and degranulation of the human mast cell line LAD2, which endogenously expresses MrgprX2, and induces production of IL-8 and CCL3. Murepavadin induces degranulation of mast cells and increases vascular permeability in the mouse abdominal cavity through MrgprX2 [162]. AG-30/5C is an angiogenic HDP that induces LAD2 human mast cells to degranulate and produce lipid mediators, including leukotriene C4, PGD2 and PGE2. Moreover, AG-30/5C increases mast cell chemotaxis and production of the cytokines GM-CSF and TNF-α and various chemokines such as IL-8, MCP-1, MCP-3, MIP-1α, and MIP-1β, and causing the phosphorylation of MAPKs and IκB [163]. In 2021, Agier et al. found that cathepsin CRAMP and cytokine IL-33 significantly promote production of Dectin-1, Dectin-2, RIG-I, and NOD1 receptors in mature mast cells and regulate their inflammatory response [166]. Dermcidin is a natural antibacterial peptide secreted by sweat glands that is usually transported to the epidermal surface by sweat. Dermcidin directly activates mast cells and induces release of the cytokines CCL1, CCL2, IL-6, and TNF-α in vitro [164].

Some peptides in the cationic pleurocidin peptide family lead to degranulation of the mast cell line LAD2, increase the mobilization of Ca^2+^ in mast cells, and induce production of the proinflammatory chemokines CCL2, CCL4, and MCP-1. Pleurocidin NCR-04 causes LAD2 to adhere, migrate, degranulate, and release cysteine, LTs and PGD2 [165].

## 6. Regulatory Effects of AMPs on DCs

### 6.1. Regulation of DCs by Mammalian-Derived AMPs

Cathelicidins inhibit TLR4- but not TLR2-mediated DC maturation and cytokine release, and this inhibition is associated with the alteration of cell membrane function and structure (Figure 5) [167]. LL-37-derived DCs display significantly upregulated endocytic capacity, modifying phagocytic receptor expression and function, upregulating costimulatory molecule expression, and enhancing secretion of the Th1 cell-inducing cytokines IL-4, IL-6, IL-12, IL-10, and TNF-α, with the promotion of Th1 cell responses in vitro [168]. LL-37 also increases the expression of HLA-DR and the costimulatory molecule CD86 in immature DCs, enhances expansion and differentiation of DCs with key characteristics of CD103^+^/CD141^+^, and upregulates CCR7 by improving migration ability [169,170]. In addition, LL-37 upregulates expression of CD40 on the surface of DCs and enhances the antigen presentation ability of DCs by promoting DC differentiation and maturation [171]. The LL-37-DNA complex can be delivered to the early endocytic chamber of plasmacytoid DCs, triggering TLR9 and type I interferon production and activating plasmacytoid DCs to develop into proinflammatory phenotypes during the onset of type I diabetes [172,173]. Self-RNA–LL37 complexes activate TLR7 and trigger secretion of IFN-α and the activation of classical myeloid DCs (MDCs) through TLR8, leading to the production of TNF-α and IL-6 and the differentiation of MDCs into mature DCs (Table 5) [174].

hBD2 and hBD3 activate plasmacytoid dendritic cells (pDCs) by enhancing the uptake of CpG and their own DNA in cells and promote DNA-induced IFN-α production in a TLR9-dependent manner [175]. hBD3 potently induces phenotypic maturation of human Langerhans cell-like DCs, including increased expression of CCR7, which mediates functional chemotactic responses to CCL19 and CCL21. hBD3-stimulated human Langerhans cell-like DCs induce strong proliferation and IFN-γ secretion by naive human T cells. hBD3 also promotes phenotypic maturation of primary human skin-migratory DCs derived from human skin explants [176]. Mouse β-defensin-2 (mDF2β) acts directly on immature DCs as an endogenous ligand of TLR-4, inducing costimulatory molecule upregulation and DC maturation and triggering the TNFR2-mediated “self-destruction” signaling cascade to kill DCs through upregulation of membrane-bound TNF-α and TNFR2 [177,178]. In addition, mouse β-defensin-14 (MBD-14) increases expression of CD40 and MHC II on the DCs surface, reduces DCs’ endocytosis ability and enhances T cell proliferation [179].

Pig-derived AMPs PMAP-23 and PMAP-36, rich in Pro and arginine, rapidly and effectively ingest nucleic acids in pDCs and subsequently produce a powerful IFN-α reaction [180].

### 6.2. Regulation of DCs by Microbial-Derived AMPs

There are many AMPs of microbial origin, but few have an immune effect on DCs. Surfactin, a bacterial cyclic lipopeptide, is produced by *B. subtilis*. Surfactin can induce DCs to release IL-6 and TNF-α, activate NF-κB signaling pathways, stimulate DC maturation, and enhance the proliferative ability of DCs [181]. Beauvericin (BEA), a mycotoxin of the enniatin family produced by toxigenic fungi, is able to activate BMDCs, manifested by increasing expression of IL-12 and CD86 [182].

### 6.3. Regulation of DCs by Other AMPs

Pep19-2.5 is a synthetic anti-lipopolysaccharide peptide. In LPS-stimulated human monocyte-derived DCs and Langerhans-like cells, Pep19-2.5 blocks IL-6 secretion, downregulates expression of maturation markers and inhibits DC migration [183].

## 7. Regulatory Effect of AMPs on Neutrophils

### 7.1. Regulation of Neutrophils by Mammalian-Derived AMPs

The human cathelicidin family member AMP LL-37 has an immunomodulatory effect on neutrophils [184]. LL-37 is a potent inhibitor of human neutrophil apoptosis and signaling pathways through P2X7R- and G-protein-coupled receptors (other than the formyl peptide receptor-like-1 molecule) (Figure 6). This process involves the modulation of Mcl-1 expression, the inhibition of BID and procaspase-3 cleavage, and the activation of phosphatidylinositol-3 kinase [185]. LL-37 promotes neutrophils to independently form neutrophil extracellular traps (NETs) and enhances the resistance of NETs to the degradation of *Staphylococcus aureus* nuclease (Table 6) [186,187]. Activated neutrophils can release NETs in response to a variety of stimuli. NETs have two important functions: capturing and destroying microorganisms including viruses, fungi, bacteria, and protozoa [188]. LL-37 enhances production of IL-8 under the control of MAPK p38 and ERK and induces phosphorylation of p38 and ERK. LL-37 stimulates ROS generation, most likely via NADPH oxidase activation and intracellular Ca^2+^ mobilization. In addition, LL-37 induces both mRNA expression and protein release of α-defensins, known as human neutrophil peptides 1–3 [189]. Finally, LL-37 significantly inhibits release of proinflammatory cytokines in neutrophils and expression of IL-1β, TNF-α, soluble trigger receptors, and damage-associated molecular patterns on triggering receptor expressed on myeloid cells-1 (TREM-1) in plasma and peritoneal fluid [190,191].

LL-37 and hBD-3 potently modulate and suppresse neutrophil apoptosis accompanied by phosphorylation of ERK-1/2, downregulation of the proapoptotic proteins tBid and BID, upregulation of the antiapoptotic proteins Bcl-x and Mcl-1, and inhibition of mitochondrial membrane potential change and caspase-3 activity; these events may occur through actions on the distinct receptors; the P2Y6 nucleotide receptor, the chemokine receptor CCR6, and the low-affinity formyl-peptide receptor FPRL1 and the nucleotide receptor P2X7, respectively [192]. hBD-1 also induces robust NET formation by targeting polymorphonuclear leukocytes [193]. In addition, hBD-2 effectively limits neutrophil infiltration in the lungs [194]. AMPs aggravates myocardial ischaemia/reperfusion injury, possibly by overactivating TLR4 signaling and the P2X7R/NLRP3 inflammasome in heart-infiltrating neutrophils, which leads to excessive secretion of IL-1β and subsequent inflammatory injury [204].

### 7.2. Regulation of Neutrophils by Insect-Derived AMPs

Scolopendrasin X and scolopendrasin IX are AMPs extracted from centipedes that increase Ca^2+^ levels in neutrophils, migrate through the G protein and phospholipase C pathways, and increase production of superoxide anions by neutrophils. Scolopendrasin X also significantly inhibits LPS-induced production of the inflammatory cytokines TNF-α and IL-6 in neutrophils. Scolopendrasin IX inhibits the inflammatory mediators TNF-α, IL-6, IL-10, and CCL2 in the joints and recruits neutrophils into the joint area [195,196].

### 7.3. Regulation of Neutrophils by Other AMPs

Cathelicidin-BF, which stimulates formation of NETs in vitro, was the first AMP to be identified in reptiles [197,198]. Cathelicidin-BF from *Bungarus auratus* significantly promotes enrichment of neutrophils in the abdominal cavity and expression of chemokines CCL2, CXCL1, and CXCL2. In vitro, it promotes secretion of chemokines by mouse macrophages in a dose-dependent manner. Cathelicidin-BF also activates the MAPK and NF-κB signaling pathways in mouse peritoneal macrophages [199].

The AMP lipocalin 2 (LCN2) from neutrophils causes chemotaxis of neutrophils by activating the ERK1/2 and MAPK p38 pathways and induces production of a variety of proinflammatory factors including IL-1α, IL-6, IL-8, and TNF-α [200]. The synthetic AMP KSLW was demonstrated to be chemotactic for neutrophils in micromolar amounts, with significantly increased F-actin polymerization in neutrophils treated with KSLW. Additionally, KSLW has been shown to inhibit oxygen radical production in PMA- and LPS-stimulated neutrophils [201]. The N-terminal peptide Ac2-26 enhances release of CXCL2 stimulated by CG/NE in neutrophils via FPR stimulation [202]. Zd-14CFR, a recombinant AMP, enhances bacterial clearance and decreases neutrophil infiltration and expression of TNF-α and IL-1β [203].

## 8. Regulatory Effect of AMPs on Eosinophils

Eosinophils are hematopoietic stem cells derived from bone marrow that have a cell-death-inducing effect on bacteria and parasites and participate in the immune response. LL-37 mediates chemotaxis of eosinophils through FPRs (Figure 7) [205]. LL-37 can act as an eosinophil-activating peptide to trigger release of inflammatory mediators and Cys-LTs from eosinophils through FPRs, enhance cytosolic phospholipase A2 activity through ERK1/2 phosphorylation, and promote release of ECP from eosinophils (Table 7) [206,207]. LL-37 significantly enhances expression of CD18 and release of IL-6, CXCL8, and CCL4 in eosinophils [208].

## 9. Application Prospects of AMPs

Peptide therapeutics have played a notable role in medicine. To date, more than 60 peptides have been approved in the US, Europe, and other major markets, and over 400 peptides have been evaluated in clinical studies [209]. Increasing attention has been given to peptides as therapeutic drugs, and it is estimated that the annual sales of peptide drugs in 2024 will be more than USD 50 billion [210]. AMPs have been proven to be promising antimicrobial agents in vitro and in vivo in laboratory testing, as well as in the fields of food preservation, food additives, and antitumor research. According to the Data Repository of Antimicrobial Peptides (DRAMP) database (http://dramp.cpu-bioinfor.org/, last updated on 4 July 2023), 96 AMPs have been developed as drug candidates for the peptide drug market, as shown in Appendix A. Twelve peptides have been approved for commercialization, including Daptomycin, Dalbavancin, Telavancin, Oritavancin, Sifuvirtide, Bacitracin, Colistin, Polymyxin B, Tyrothricin, Vancomycin, Gramicidin S, and Gramicidin D. Eighty peptides are under clinical development, with 50 in clinical trials and 30 in preclinical stages. Three AMPs, MSI-78, Iseganan, and rBPI21, failed in phase III clinical trials. CZEN-002 (Melantropin, (CKPV)2), EA-230, RDP58 (Delmitide), Ghrelin, and EA360 are used in the treatment of the inflammatory diseases.

As mentioned above, the demand for AMPs is continuously increasing, and an increasing number of AMPs are entering clinical trials. However, there are numerous problems that limit the breadth and depth of their clinical application. (1) Due to the presence of AMPs in extremely small amounts in organisms, extracting them from natural sources involves a complex process with low yield, high time consumption, and high expense, resulting in a low extraction rate [211,212,213]. (2) Moreover, many AMPs are easily decomposed by proteolytic enzymes in vivo and quickly excreted through the kidneys. The particularly short half-life results in low antibacterial activity [214]. (3) AMPs have low physical and physiological stabilities, and their activities are sensitive to environmental conditions such as pH, temperature, ions, and serum factors, resulting in unstable effects [215]. (4) Toxicity is also an important factor hindering the clinical application of AMPs [214]. It has been reported that mammalian cell membranes may become targets for natural AMPs [216]. (5) AMPs also eliminate beneficial bacterial communities in vivo due to their broad-spectrum antibacterial properties [217]. (6) Most AMPs are limited to local administration or the intravenous injection route.

How to solve these problems is a direction that needs to be studied in the future. Large-scale production of AMPs can be achieved through application of biotechnological engineering and fermentation. A structural optimization strategy with chemical synthesis was implemented to overcome the potential toxicity to animal cells and low metabolic stability. In recent years, our group has had a strong ongoing interest in structural modification of PMAPs and has obtained novel modified peptides with safer effects, increased stability and higher antimicrobial activity [218,219,220,221,222,223,224,225,226]. In the future, artificial intelligence should be applied for structural optimization of AMPs to obtain better novel active analogs.

## 10. Summary and Outlook

AMPs represent the first line of defense against pathogens in the innate immune system. Some AMPs are produced and secreted by the secondary granules of white blood cells (mainly neutrophils), and their concentration increases rapidly at the site of inflammation. Interestingly, AMPs in turn influence numerous biological processes of immune cells ranging from modulation of inflammatory responses to promotion of wound healing [227]. This review summarized the roles of AMPs from different sources in the regulation of immune cell function.

(1) At present, most of the research on the anti-inflammatory response focuses on macrophages, whereas research into other immune cells is limited, and NK cells and basophils have not been studied at all. The limited research into the effect of different kinds of AMPs on immune cells may be one of the things affecting the clinical use of AMPs. Although the individual effects of AMPs on the anti-inflammatory functions of immune cells, including by direct action on the cells or by influencing inflammatory mediators, signaling pathways, and expression of genes related to apoptosis, were outlined in this review, it is impossible to generalize the most common points as exhibiting regularity among various types of immune cells based on the current research.

(2) There are many signals that cause the body to produce inflammatory reactions, such as biological factors (bacteria, fungi, viruses, etc.), physical factors (high temperature, ultraviolet light, mechanical damage, etc.), and chemical factors (strong acids, strong bases, etc.). As mentioned above, most studies to date have focused on inflammation induced by LPS from Gram-negative bacteria. For example, CAP11, studied by Shibusawa K et al. in 2009 [16], β-defensin 2, studied by Huang C et al. in 2017 and 2022 [26,27], and hBD130, studied by D et al. in 2022 [29], all achieved the goal of suppressing inflammation by inhibiting LPS. However, few studies have investigated the anti-inflammatory role of AMPs in immune cells following stimulation with LTA from Gram-positive bacteria, fungi, and viruses, as well as by physical and chemical factors.

(3) Regarding the inflammatory environment, most studies on the contribution of AMPs to the function of immune cells have included in vitro experiments, but few in vivo studies have been performed. In the physiological environment, it is unknown whether AMPs have an anti-inflammatory effect similar to that of immune cells in vitro, which is one of the important factors making it difficult to apply AMPs in clinical practice.

(4) The regulatory effect of AMPs from different sources on the functions of immune cells cannot be uniformly described. Different types of AMPs have specific antimicrobial mechanisms. Molecular weight, amino acid composition, and charge affect AMP antibacterial functions. However, there are few articles pointing out the different effects of different types of AMPs on immune cell function.

In summary, AMPs are an evolutionarily conserved component of the innate immune system and exhibit antibacterial, antiviral, antifungal, antiparasitic, antitumor, and immunomodulatory properties. This review clarifies the findings of current research on the mechanism of AMP-mediated regulation of immune cell function. These findings may be useful for future researchers aiming to conduct a series of in-depth investigations on the multifunctional properties of AMPs. However, it is necessary to determine whether AMPs from different sources have different effects on the function of immune cells. More rapid and efficient elucidation of the anti-inflammatory mechanisms of AMPs must be conducted in different kinds of immune cells. Whether AMPs can inhibit inflammatory reactions caused by LPS is one of the important factors regarding whether they can be applied in clinical practice in the future. Further research will be conducted investigating the specific mechanisms by which AMPs regulate immune function in physiological environments, as well as their toxic effects on host cells. With the development of biomedicine and related technologies, an increasing number of AMPs will gradually enter clinical application.

## Figures and Tables

**Figure 1 pharmaceutics-15-02278-f001:**
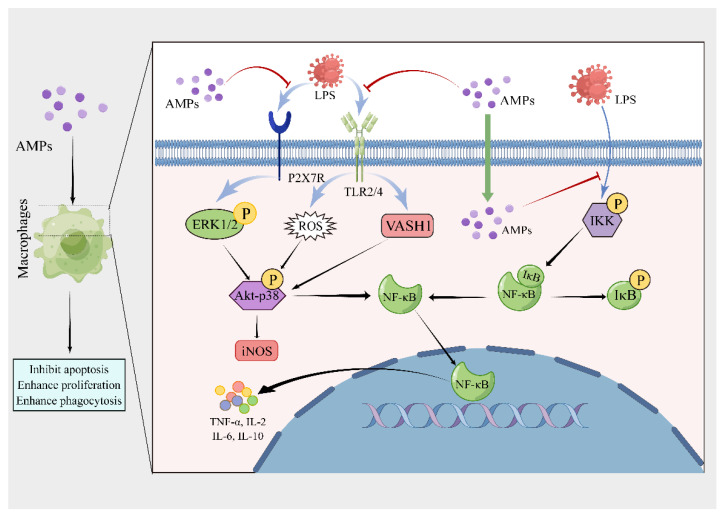
The regulatory mechanism of AMPs on macrophages (Drew in Figdraw.). AMPs inhibit apoptosis of macrophages, and enhance proliferation and phagocytosis of macrophages. AMPs inhibit LPS-induced inflammatory responses through P2X7R receptors or TLR2/4 receptors. AMPs inhibit Akt-p38 phosphorylation and NF-κB activation in different pathways induced by LPS through different receptors. Moreover, AMPs enter macrophages directly to block IKK phosphorylation, inhibit NF-κB pathway activation and production of the inflammatory factors.

**Figure 2 pharmaceutics-15-02278-f002:**
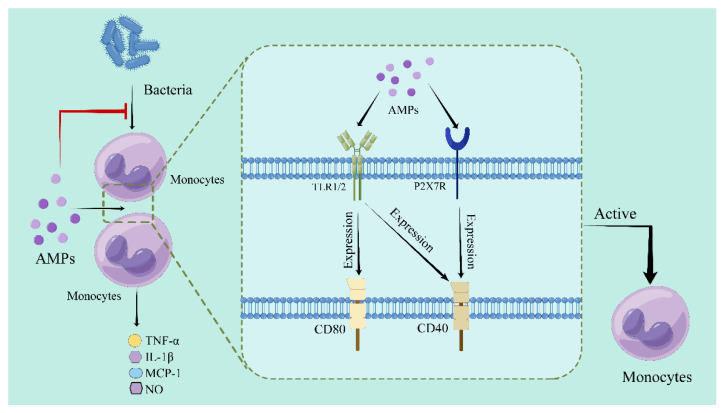
The regulatory mechanism of AMPs on monocytes (Drew in Figdraw.). AMPs act on the TLR1/2 receptor and R2X7R receptor on the surface of monocytes, causing expression of CD80 and CD40 and activation of monocytes. AMPs inhibit production of the inflammatory mediators in monocytes after treatment with bacteria.

**Figure 3 pharmaceutics-15-02278-f003:**
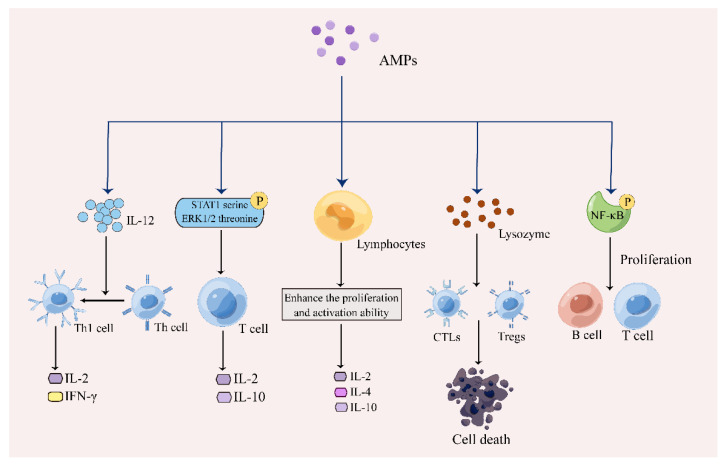
The regulatory mechanism of AMPs on lymphocytes (Drew in Figdraw.). AMPs induce apoptosis of CTLs and Tregs by inducing the release of granzyme. AMPs promote proliferation of T and B cells through NF-κB activation, and also directly enhance proliferation and activation of lymphocytes, and production of IL-12, IL-2, and IFN-γ by promoting transformation of Th cells into Th1 cells. T cells were stimulated by AMPs to produce IL-2/10 by inducing STAT1 serine phosphorylation and ERK1/2 threonine phosphorylation.

**Figure 4 pharmaceutics-15-02278-f004:**
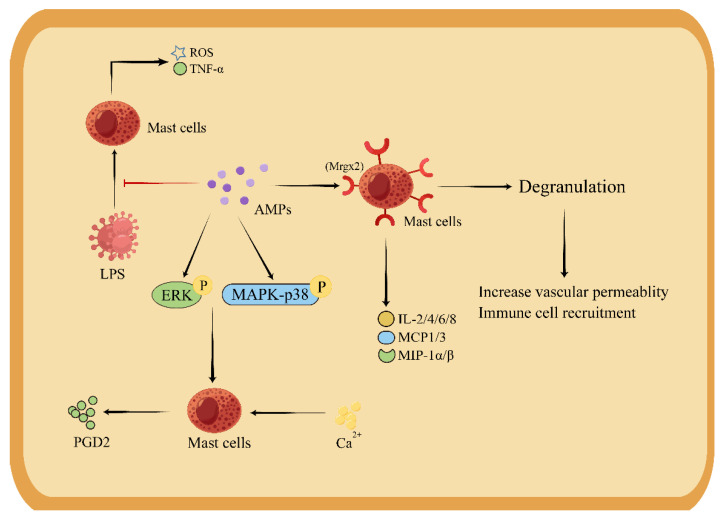
The regulatory mechanism of AMPs on mast cells (Drew in Figdraw.). AMPs induce an increase in Ca2+ concentration in mast cells through ERK and MAPK-p38 phosphorylation. By acting on the Mrgx2 receptor, mast cells release inflammatory mediators and cause degranulation, thus increasing vascular permeability, and recruit immune cells. AMPs directly inhibit production of ROS and TNF-α in mast cells induced by LPS.

**Figure 5 pharmaceutics-15-02278-f005:**
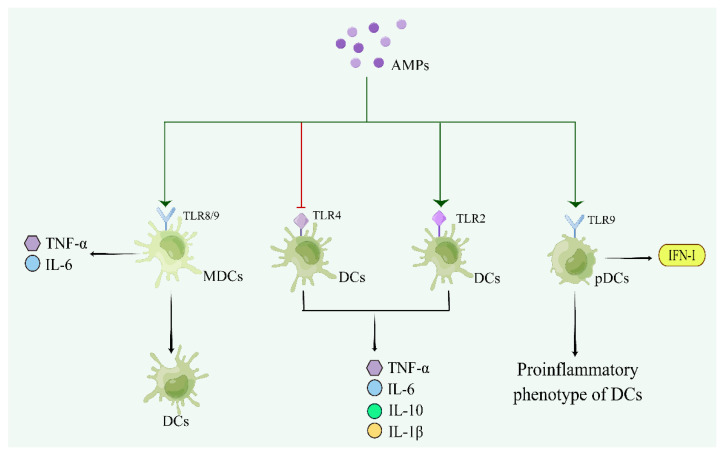
The regulatory mechanism of AMPs on dendritic cells (Drew in Figdraw.). AMPs promote cytokine production by acting on TLR2 rather than TLR4 receptor on the surface of DC cells, and enhance production of INF-I by acting on the TLR9 receptor on the surface of MDC cells and then transform into a pro-inflammatory phenotype of DC cells. AMPs convert MDCs into DCs via TLR9, and produce TNF-α and IL-6.

**Figure 6 pharmaceutics-15-02278-f006:**
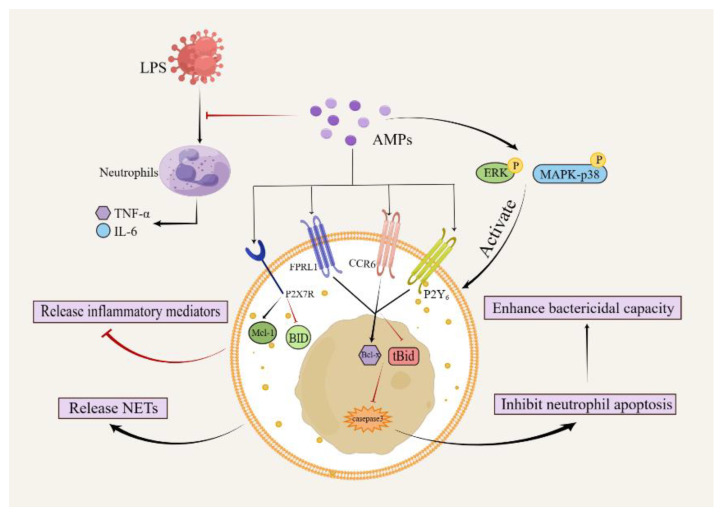
The regulatory mechanism of AMPs on neutrophils (Drew in Figdraw.). AMPs promote production of apoptotic proteins Mcl-1 and Bcl-x, inhibit production of pro-apoptotic proteins BID and tBid, caspase-3 activation and neutrophil apoptosis, and enhance bactericidal ability through P2X7R, FPRL1, CCR6, and P2Y6 receptors. AMPs directly inhibit release of TNF-α and IL-6 in neutrophils induced by LPS. AMPs activate neutrophils to release NETs and inhibit release of inflammatory mediators by activating ERK and MAPK pathways.

**Figure 7 pharmaceutics-15-02278-f007:**
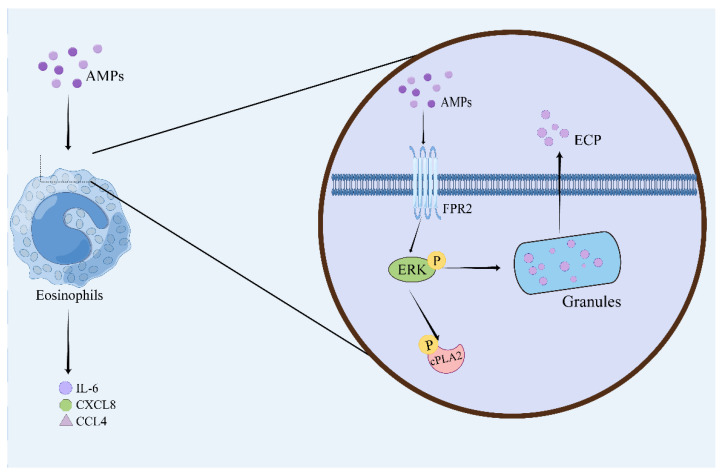
The regulatory mechanism of AMPs on eosinophils. (Drew in Figdraw.). AMPs induce ERK phosphorylation through FPR2 receptors, enhance cPLA2 activity, and promote release of ECP, IL-6, and CXCL4/8 from eosinophils.

**Table 1 pharmaceutics-15-02278-t001:** Outline of research on the regulatory effects of AMPs on macrophages in recent years.

Peptide Name	Source	Amino Acid Sequence	Inflammatory Mediator	Signaling Pathway	Functions	Ref.
BSN-37	Bovine	FRPPIRRPPIRPPFYPPFRPPIRPPIFPPIRPPFRPP	IL-2, IFN-γ, IL-4, and IL-10 ↑	-	Promote expression of CD40/80, MHC Ⅰ and MHC Ⅱ	[13]
PR-39	Porcine	RRRPRPPYLPRPRPPPFFPPRLPPRIPPGFPPRFPPRFP	-	-	Inhibit apoptosis and reduce caspase-3 activity, enhance the ability to kill and clear intracellular bacteria	[14,15]
CAP11	*Guinea pig*	GLRKKFRKTRKRIQKLGRKIGKTGRKVWKAWREYGQIPYPCRI	-	-	Inhibit the binding of LPS, release of HMGB1, and the death of necrotic cells	[16]
LL-37	Human	LLGDFFRKSKEKIGKEFKRIVQRIKDFLRNLVPRTES	IL-8, MCP-1, LTB4, and COX-2 ↑; MIP-1α, IL-12, and IL-1β ↓	Promote ERK1/2 and MAPK p38	Promote proliferation and phagocytosis, neutralize LPS and inhibit the response of P2X7R to ATP, inhibit activation of caspase-1 and pyroptosis	[17,18,19,20,21,22,23]
HBD2	Human	GIGDPVTCLKSGAICHPVFCPRRYKQIGTCGLPGTKCCKKP	INF-β, IFN-γ, IL-6, IL-1β, and TNF-α ↑	-	-	[24]
HBD3	Human	GIINTLQKYYCRVRGGRCAVLSCLPKEEQIGKCSTRGRKCCRRKK	TNF-α and IL-6 ↓	Inhibit MAPK p38 and ERK1/2	-	[25]
PBD-2	Porcine	DHYICAKKGGTCNFSPCPLFNRIEGTCYSGKAKCCIR	IL-12, IL-6, IL-1β, TNF-α, and NO ↓	Inhibit NF-κB and Akt	-	[26,27]
DEFB123	Human	GTQRCWNLYGKCRYRCSKKERVYVYCINNKMCCVKPK	TNF-α ↓	Inhibit MAPK	-	[28]
HBD130	Human	GVIPGQKQCIALKGVCRDKLCSTLDDTIGICNEGKKCCRRWWILEPYPTPVPKGKSP	NO, IL-1β, IL-6, and TNF-α ↓	-	-	[29]
Indolicidin	Bovine	ILPWKWPWWPWRR-NH2	NO and iNOS ↓	-	-	[30]
JH-3	Bovine	RRFKLLSHSLLVTLASHL	IL-2, IL-6, TNF-α, LDH, and TNF-Rα2 ↓	Inhibit MAPK p38	Inhibit activation of caspase-3 and caspase-8, reduce macrophages apoptosis induced by *Salmonella*, inhibit release of cytochrome C in the cytoplasm and the expression of caspase-8/9	[31,32]
BMAP-18	Bovine	GRFKRFRKKFKKLFKKLS-NH2	NO and TNF-α ↓	-	-	[33]
Bac5	Bovine	RFRPPIRRPPIRPPFYPPFRPPIRPPIFPPIRPPFRPPLGPF	TNF-α ↑	-	Activate macrophage-like THP-1 cells	[34]
Bovine lactoferricin	Bovine	NH2PHELYSCYSARGARGTRPGLNTRPARGMETLYSLYSLEUGLYALAPROSERILETHRCYSVALARGARGALAPHECOOH	TNF-α, IL-6, NO, and iNOS ↓	Inhibit NF-κB and MAPK	-	[35]
HLF1-11	Human	GRRRRSVQWCA	IL-10 ↑	-	Promote monocyte differentiation into macrophages driven by GM-CSF	[36]
HEPC1	Musculus	QSHLSMCRYCCNCCRNNKGCGFCCKF	-	-	Promote iron storage and inhibit iron release of Ana.1 and RAW264.7, and promote iron retention	[37]
HEPC2	Musculus	NPAGCRFCCGCCPNMIGCGVCCRF	-	-	Promote iron storage and inhibit iron release of Ana.1 and RAW264.7, and promote iron retention	[38]
CHensinin-1b	Frog	SKVWRHWRRFWHRAHRKL	TNF-α, IL-6, IL-1β, and NO ↓; IL-10 and TGF-β1 ↑	Inhibit NF-κB and MAPK	-	[39]
W3R6	Frog	VWRRWRRFWRR	TNF-α, IL-6, IL-1β, and NO ↓; IL-10 and TGF-β1 ↑	Inhibit NF-κB and MAPK	-	[39]
Temporin-1CEa	Frog	FVDLKKIANIINSIFGK-NH2	TNF-α and IL-6 ↓	Inhibit NF-κB and MAPK	-	[40]
LK2 (6)	Frog	FVKLKKIANIINSIFKK-NH2	TNF-α and IL-6 ↓	Inhibit NF-κB and MAPK	-	[40]
LK2 (6) A (L)	Frog	FVKLKKILNIINSIFKK-NH2	TNF-α and IL-6 ↓	Inhibit NF-κB and MAPK	-	[40]
PN-CATH1	Frog	KKCNFFCKLKKKVKSVGSRNLIGSATHHHRIYRV	IL-6, IL-1β, and TNF-α ↓	-	-	[41]
PN-CATH2	Frog	EGCNILCLLKRKVKAVKNVVKNVVKSVVG	IL-6, IL-1β, and TNF-α ↓	-	-	[41]
Temporin-1TI	Frog	FVQWFSKFLGRIL-NH2	TNF-α, NO, and iNOS ↓	-	-	[42]
OL-CATH2	Frog	RKCNFLCKVKNKLKSVGSKSLIGSATHHGIYRV	TNF-α, IL-1β, and IL-6 ↓	-	-	[43]
HR-CATH	Frog	ASKKGKCNLLCKLKQKLRSVGAGTHIGSVVLKG	-	-	Induce macrophage chemotaxis and enhance respiratory burst	[44]
Nv-CATH	Frog	NCNFLCKVKQRLRSVSSSHIGMAIPRPRG	NO, IL-6, TNF-α, and IL-1β ↓; CXCL1/2 and CCL2 ↑	Inhibit NF-κB-NLRP3 and MAPK	-	[45]
TK-CATH	Salamander	GGQDTGKEGETGKKKKSDNWFMNLLNKFLELIGLKEAGDDSEPFCFTCIFDMFSQ	TNF-α, IL-6, and MCP-1 ↓	Inhibit MAPK p38	-	[46]
CLP-19	*Limulus*	CRKPTFRRLKWKIKFKFKC	TNF-α ↓	-	-	[47]
Mt6	Housefly	KKFKKTAKWLIKSAWLLLKSLALKMK	IL-1β and TNF-α ↓	Inhibit MAPK	-	[48]
D-Mt6	Housefly	KKFKKTAKWLIKSAWLLLKSLALKMK	IL-1β and TNF-α ↓	Inhibit MAPK	-	[48]
HI-3	*Hermetia illucens*	-	IL-6, TNF-α, IL-1β, and NO ↓; IL-10 ↑	Inhibit NF-κB	Enhance the phagocytosis and inhibit LPS-induced differentiation of RAW264.7, increase the superoxide dismutase activity and total antioxidant capacit	[49,50]
CEMA	Insect	KWKLFKKIGIGAVLKVLTTGLPALKLTK	IL-1β, IL-6, TNF-α, MIP-1α, and MIP-1β ↓	-	Induce expression of gene-involved in cell adhesion and apoptosis	[51]
Peptide A	Beetle	RLYLRIGRR-NH2	TNF-α and NO ↓	Inhibit NF-κB	-	[52]
Peptide B	Beetle	RLRLRIGRR-NH2	TNF-α ↓	-	-	[52]
Defensin 1	Beetle	YPLDQVEEQDEHQVAHIRVRRVTCDLLSAEAKGVKVNHAACAAHCLLKRKRGGYCNKRRICVCRN	IL-1β, IL-6, IL-8, IL-10, IL-12p70, IL-23, and TNF-α ↓	-	-	[53]
CopA5	*Dung beetle*	LLCIA	NO and TNF-α ↓	Inhibit STAT1	Inhibit the phagocytic activity of PEM	[54]
ToAP3	*Brazilian scorpion*	FIGMIPGLIGGLISAIK	TNF-α and IL-1β ↓; IL-10 ↑	-	Decrease co-stimulatory molecules (CD80 and CD86)	[55]
ToAP4	*Brazilian scorpion*	MQIKHLITLFFLVLIVADQCSAFFSLIPSLIGGLVSAIKGGRRKREIAAQIEQYRDLQKREAELEELLDRLPMF	TNF-α and IL-1β ↓; IL-10 ↑	-	Increase expression of MHC Ⅱ	[55]
FA1	Scorpion	-	IL-10, IL-12p70, and TNF-α ↑	-	Activate phagocytic activity	[56]
CM4	Bombyx	RWKIFKKIEKVGQNIRDGIVKAGPAVAVVGQAATI	TNF-α, NO, and IL-6 ↓	-	-	[57]
Cecropin DH	Chinese oak silk moth	KWKIFKKIEKVGRNIRNGIIKAVAVLGEAKAL	NO and TNF-α ↓	-	-	[58]
Papiliocin	*Swallowtail butterfly*	RWKIFKKIEKVGRNVRDGIIKAGPAVAVVGQAATVVK-NH2	NO, TNF-α, and MIP-2 ↓	Inhibit NF-κB and TLR4	-	[59]
Periplanetasin-5	Cockroach	MKTFLRLYRSLINKVLH	TNF-α and IL-6 ↓	Inhibit NF-κB and MAPKs	-	[60]
AK-N’	Spider	FKGLAKLLKIGLKALAKVIQ	IL-6, IL-1β, and TNF-α ↓	-	-	[61]
AK-N’ m	Spider	NKGLAKLLKIGLKALESVIQ	IL-6, IL-1β, and TNF-α ↓	Inhibit TLR4	-	[61]
KN-17	*Tianchan pupa*	KWKVFKKIEKMGRNIRN	iNOS, TNF-α, and IL-1α ↓; Arg1 and TGF-β ↑	Inhibit NF-κB	Enhance RAW264.7 to transform from M1 to M2	[62]
PN5	*Pine needle*	FKFLARTGKFL	IL-6 and TNF-α ↓	Inhibit NF-κB and MAPKs	-	[63]
Sublancin	*Bacillus subtilis* 168	GLGKAQCAALWLQCASGGTIGCGGGAVACQNYRQFCR	IL-1β, IL-6, TNF-α, and NO ↑	Promote NF-κB, MAPK, and TLR4	Enhance the phagocytosis and killing activity of RAW264.7 and mouse peritoneal macrophages against methicillin-resistant *Staphylococcus aureus*	[64]
CSP32	*Bacillus* spp.	APLEXXIFHDN	NO, TNF-α, IL-1β, MCP-1, and PGE2 ↑	Promote NF-κB and MAPK	Stimulate phagocytosis, induce the appearance of M1 type macrophages, increase the number of Ca^2+^ positive macrophages, upregulate phospholipase C and activate protein kinase Cε	[65]
MS15	*Bacillus*	-	ROS and NO ↓	-	Increase translation and transcription levels of catalase, glutathione peroxidase, and superoxide dismutase	[66]
Macrolactin	*Bacillus subtilis*	-	iNOS, COX-2 and IL-6 ↓	-	-	[67,68]
Surfactin	*Bacillus subtilis*	ELIVDIL	-	Inhibit NF-κB, p38, JNK and Akt	Impair the antigen delivery function of macrophages	[69]
OMS A	Marine bacteria	VVTTVLVVWVFV	TNF-α, IL-1β, CCL5, IL-12p40, and iNOS ↑	-	Downregulate arginase-1 expression	[70]
BacSp222	*Staphylococcus*	-	NO, TNF-α, IFN-β, IL-1α, IL-10, IL-27 and MCP-1 ↑	Promote NF-κB	-	[71]
BEA	Fungi	-	NO ↓	Inhibit NF-κB	-	[72]
EnnB	Fungi	-	IL-1β ↑	-	Caused a G0/G1-arrest, M2-like macrophage differentiation, apoptosis and necrosis	[73]
CATH-2	Chicken	RFGRFLRKIRRFRPKVTITIQGSARF-NH2	IL-6 and IL-1β ↓	Inhibit TLR2 and TLR4	-	[74]
CATH-B1	Chicken	PITYLDAILAAVRLLNQRISGPCILRLREAQPRPGWVGTLQRRREVSFLVEDGPCPPGVDCRSCEPGALQHCVGTVSIEQ	IFN-β, IL-1β, IL-6, and IL-8 ↓; IL-10 ↑	Inhibit TLR4	Enhance phagocytosis	[75]
Fowlicidin-1(6–26)	Chicken	WPLVIRTVIAGYNLYRAIKKK-NH2	IL-1β, CCL2, and CCL3 ↑	-	Enhance the surface expression of MHC Ⅱ and CD86 on RAW264.7	[76]
CNK-2	Chicken	RRQRSICKQLLKKLRQQLSDALQNNDD	IL-1β ↓; CCL4 ↑	Promote MAPK	-	[77]
DCATH 12-4	Duck	LIKKIYRKWKRW-NH2	NO, TNF-α, and iNOS ↓	-	-	[78]
DCATH 12-5	Duck	LWKKIYRKWKRW-NH2	NO, TNF-α, and iNOS ↓	-	-	[78]
AvBD8	Chicken	MKILYFLLAVLLTVLQSSLGFMRVPNNEAQCEQAGGICSKDHCFHLHTRAFGHCQRGVPCCRTVYD	IL-1β, INF-γ, IL-12p40, CCL4, CXCL13, and CCL20 ↑	Promote MAPK	-	[79]
Hep20	*Rainbow trout*	ICIFCCGCCHRSKCGMCCKT	IL-10, IL-1β, and TNF-α ↑	-	-	[80]
TP4	*Nile tilapia*	H-FIHHIIGGLFSAGKAIHRLIRRRRR-OH	NO, TNF-α, IL-1β, and IL-6 ↓	Promote MAPK, ERK, and IL-10-STAT3	Enrich markers of M2 macrophages	[81]
EPI	*Orange-spotted grouper*	H-GFIFHIIKGLFHAGKMIHGLV-OH	IL-6, COX-2, iNOS, TNF-α, and ROS ↓	Inhibit Akt and NF-κB	Reduce the cytotoxicity of RAW264.7 induced by LPS	[82]
Epi-1	*Orange spotted grouper*	GFIFHIIKGLFHAGKMIHGLV	TNF-α, IL-6, and IL-1β ↓; TGF-β and Sytx1 ↑	-	Induce expression of MHC related genes	[83]
Scolopendrasin VII	*Scolopendra subspinipes mutilans*	FCTCNVKGFNAKNKRGIIYP-NH2	-	Promote ERK and Akt	Stimulate actin polymerization and the chemotactic migration of macrophages	[84]
Pardaxin	Marine fish species	GFFALIPKIISSPLFKTLLSAVGSALSSSGGQE	-	Promote MyD88	Induce THP-1 and U937 cells to differentiate into macrophages with phagocytic ability, increase expression of MyD88, and reduce the phagocytic ability and superoxide anion production of leukemia cells	[85]
LjP-3	Japanese sea bass	FFGMLIHGAIHAGKVIHJLIHG	IL-1β, TNF-α, and TGF-β ↓	-	Promote macrophage chemotaxis and phagocytosis	[86]
LjP-2	Japanese sea bass	FLKSIWRAAKGAIRGAKSGWRA	IL-1β, TNF-α, and TGF-β ↓	-	Promote macrophage chemotaxis and phagocytosis	[87]
NKHs27	*Sevenband grouper*	KLTSKLKSICDQIGLLKALCRKSVKTH	-	-	Enhance the respiratory burst and upregulate immune-related genes expression	[87]
CNMs	Synthesis	-	TNF-α, IL-6, IL-8, IL-1β, and NO ↓	Inhibit NF-κB, MAPK, and TLR4	-	[88]
LF-KR	Synthesis	RRWQWRPKRIVKLIKKWLR-NH2	NO and TNF-α ↓	-	-	[89]
PapMA	Synthesis	RWKIFKKIPKFLHSAKKF-NH2	NO, TNF-α, IL-6, IL-1β, and MIP-1/2 ↓	-	-	[90]
KR-1	Synthesis	KKKKKKRAFARWRAFAR	iNOS, TNF-α, and IL-1β ↓	Inhibit NF-κB	Increase the percentage of M2 phenotype in macrophages	[91]
KR-2	Synthesis	KKKKKKRRFRRWRRFRR	iNOS, TNF-α, and IL-1β ↓	Inhibit NF-κB	Increase the percentage of M2 phenotype in macrophages	[91]
P18	Synthesis	KWKLFKKIPKFLHLAKKF-NH2	iNOS, IL-1β, TNF-α, and NO↓	-	-	[92]
P18-W6	Synthesis	KKKLFWKIPKFLHLAKKF-NH2	NO ↓	-	-	[93]
P18-W8	Synthesis	KIKLFKKWPKFLHLAKKF-NH2	NO ↓	-	-	[93]
P18-Nala^9^	Synthesis	KWKLFKKIaKFLHLAKKF-NH2	NO ↓	-	-	[93]
GL13K	Synthesis	GL13KGKIIKLKASLKLLCONH2	M1: TNF-α and IL-1β ↓; M2: IL-10 and TGF-β3 ↑	-	Inhibit proliferation of M1 type macrophages	[94,95]
KLW-f	Synthesis	KWKKLLKKfLKKLKKLLK-NH2	NO ↓	-	-	[96]
KLK	Synthesis	KLKLLLLLKLK	NO, TNF-α, iNOS, COX-2, IL-1β, and PGE2 ↓	Inhibit NF-κB	-	[97]
Kn2-7	Synthesis	FIKRIARLLRKIF	IL-10, TNF-α ↑	-	Enhance the uptake of CpG DNA	[98]
K_9_L_8_W	Synthesis	KLKKLLKKWLKLLKKLLK-NH2	NO, TNF-α, and iNOS ↓	-	-	[99]
Lf6-pP	Synthesis	RRWQWRpPRWQWRR-NH2	TNF-α and IL-6 ↓	-	-	[100]
KLA-NNPs	Synthesis	KLAKLAKKLAKLAK	-	-	Decrease phagocytosis and reduce the activity of intracellular bacteria and caspase-1	[101]
WK2	Synthesis	(WK)_2_CTKSGC(KW)_2_	iNOS, IL-8, IL-4, IL-1β, and TNF-α ↓	-	-	[102]
LL37-C15	Synthesis	AGEDPHGYFLPGQFA-GG-LLGDFFRKSKEKIGKEFKRIVQRIKDFLRNLVPRTES	TNF-α, IL-1β, iNOS, and IL-6 ↓	-	-	[103]
CC34	Synthesis	GWLKKIGKKIERVGQHTRDAILPILSLIGGLLGK	TNF-α, IL-1β, IL-6, ROS, iNOS, and COX-2 ↓	Inhibit NF-κB	-	[104]
GW-A2	Synthesis	GAKYAKIIYNYLKKIANALW	NO, iNOS, COX-2, TNF-α, and IL-6 ↓	Inhibit NF-κB and MAPK	-	[105]
HPRP-A1/A2	Synthesis	Ac-FKKLKKLFSKLWNWK-amide 15 mer	-	-	Reduce the survival ability of tachyzoites and adherence and invasion in macrophages, disrupt the integrity of the tachyzoite membrane	[106]
Cbf-14	Cathelin-like domain	RLLLRKFFRKLKKSV	ROS, NO, and iNOS ↓	Inhibit NF-κB, PI3K, and MAPK	-	[107]
WALK11.3	Antimicrobial model peptide	LKWLKKLLKKL-NH2	NO, COX-2, IL-1β, IL-6, INF-β, and TNF-α ↓	Inhibit TLR4	-	[108]
WALK244.04	Antimicrobial model peptide	LLKWLKKKWLK-NH2	iNOS, COX-2, and IL-1β ↓	Inhibit TRIF	-	[109]
WALK243.04	Antimicrobial model peptide	LLKWLKKWL-NH2	iNOS, COX-2, and IL-1β ↓	Inhibit TRIF	-	[109]
KAMPs	Keratin 6a	-	IL-6, TNF-α, CXCL1, and CXCL10 ↓	Inhibit NF-κB, IRF3	Reduce cell surface availability of TLR2 and TLR4	[110]

-: No description found. ↑: Increased levels. ↓: Reduced level.

**Table 2 pharmaceutics-15-02278-t002:** Outline of research on the regulatory effect of AMPs on monocytes in recent years.

Peptide Name	Source	Amino Acid Sequence	Inflammatory Mediator	Signaling Pathway	Functions	Ref.
LL-37	Human	LLGDFFRKSKEKIGKEFKRIVQRIKDFLRNLVPRTES	IFN-γ ↑	-	-	[122,123,124]
HBD1	Human	DHYNCVSSGGQCLYSACPIFTKIQGTCYRGKAKCCK	-	-	Promote differentiation of monocyte derived from Cord blood of human newborns into immature DCs and maturation of monocyte sderived DCs	[126]
HBD3	Human	GIINTLQKYYCRVRGGRCAVLSCLPKEEQIGKCSTRGRKCCRRKK	-	Promote TLR1/2 and MyD88	Induce activation of monocyte and expression of CD80, CD86, and CD40 on monocytes	[127,128]
LEAP-2	*Boleophthamus pectinirostris*	MTPLWRILNSKPFGAYCQNNYECSTGLCRAGFCATMHRSATVSVTN	TNF-α and IL-1β ↓	-	Enhance respiratory burst and bactericidal ability	[129]
BpNKLP40	*Boleophthamus pectinirostris*	SIKAKLLAVCKNIGLLKSLCQKFVNKHLGVLIEELTTTDD	TNF-α, IL-1β, and IFN-γ ↑ TGF-β and IL-10 ↓	-	-	[130]
CXCL14-C17-a2	Synthesis	KRFIKWYKAWNKKWRKY-NH2	NO, TNF-α, IL-6, and MCP-1 ↓	-	-	[131]
CXCL14-C17-a3	Synthesis	KRFKKWYKAWRKKWRKY-NH2	NO, TNF-α, IL-6, and MCP-1 ↓	-	-	[131]

-: No description found. ↑: Increased levels. ↓: Reduced level.

**Table 3 pharmaceutics-15-02278-t003:** Outline of research on the regulatory effect of AMPs on lymphocytes in recent years.

Peptide Name	Source	Amino Acid Sequence	Inflammatory Mediator	Signaling Pathway	Functions	Ref.
LL37	Human	LLGDFFRKSKEKIGKEFKRIVQRIKDFLRNLVPRTES	-	-	Enhance delivery of CTLs and Tregs cells and induce apoptosis	[132,133,134]
HBD2	Human	GIGDPVTCLKSGAICHPVFCPRRYKQIGTCGLPGTKCCKKP	-	-	Promote CD4^+^ T cell proliferation	[135]
HBD3	Human	GIINTLQKYYCRVRGGRCAVLSCLPKEEQIGKCSTRGRKCCRRKK	IL-2 and IL-10 ↑	Inhibit STAT1	Promote CD4^+^ T cells proliferation, activate T cells and enhance T cells effect function	[135,136]
BSN-37	Bovine	FRPPIRRPPIRPPFYPPFRPPIRPPIFPPIRPPFRPP	IL-2, IL-10, and IL-6 ↑	-	Enhance proliferation and activation ability of mouse spleen lymphocytes	[13]
Frenatin 2.1S	Loach	GLVGTLLGHIGKAILG-NH2	-	-	Enhance activation and homing ability of Th1 cells and NKT cell in the abdominal cavity of mice and expression of MHC II molecules on macrophages, increase the percentage of M1 macrophages	[140]
Cecropin	Tianchan pupa	GWLLKLGKRIERIGQHTRDATIQGLGIAQQAANVAATAR-NH2	-	-	Increase the number of lymphocytes in the jejunal epithelium	[141,142,143]
Immunopeptide	*Silkworm chrysalis*	DHAV	IL-12 and IL-6 ↑	Promote NF-κB	Promote the transformation of mouse Th cells to Th1 type	[144]
Sublancin	*Bacillus subtilis*	GLGKAQCAALWLQCASGGTIGCGGGAVACQNYRQFCR	-	-	Increase the ratio of CD4^+^/CD8^+^ in splenocyte	[145]
Taikanglibao	Synthesis	-	-	-	Improve the spleen index of weaned piglets, promote lymphocyte proliferation and reduce lymphocyte apoptosis	[146]
LF-6	Recombination	KWRQWQSKWRRTNPWFWIRR	IL-1, IL-2 and IL-6 ↓	-	-	[147]

-: No description found. ↑: Increased levels. ↓: Reduced level.

**Table 4 pharmaceutics-15-02278-t004:** Outline of research on the regulatory effect of AMPs on mast cells in recent years.

Peptide Name	Source	Amino Acid Sequence	Inflammatory Mediator	Signaling Pathway	Functions	Ref.
LL-37	Human	LLGDFFRKSKEKIGKEFKRIVQRIKDFLRNLVPRTES	IL-4, IL-5, and IL-1β ↑	Promote MAPKs, P13K, and Akt	Activate MrgX2-induced degranulation of human mast cells and release of de novo synthesized mediators	[152,153,154,155]
HBD2	Human	GIGDPVTCLKSGAICHPVFCPRRYKQIGTCGLPGTKCCKKP	Histamine and PGD2 ↑	-	Mobilize intracellular Ca^2+^ and release histamine	[157]
HBD3	Human	GIINTLQKYYCRVRGGRCAVLSCLPKEEQIGKCSTRGRKCCRRKK	PGD2 ↑	Promote MAPK p38 and EPK1/2	Increase intracellular Ca^2+^ concentration and degranulation	[158]
HBD4	Human	EFELDRICGYGTARCRKKCRSQEYRIGRCPNTYACCLRKWDESLLNRTKP	PGD2 ↑	Promote MAPK p38 and EPK1/2	Increase intracellular Ca^2+^ concentration and degranulation. Enhance vascular permeability	[158]
Brevinin-2KP	Frog	MFTMKKSLLLLFFLGTVSLSLCEQERGADEDDGGEMTEELKRGVITDALKGAAKTVAAELLKKAHCKLTNSC	Histamine ↑	-	Promote degranulation and histamine release of mast cells	[159]
Api88	Honeybee	Gu-ONNRPVYIPRPRPPHPRL-NH2	TNF-α ↓	-	Trigger degranulation and intracellular Ca^2+^ mobilization in human MC	[160]
AMP-IBP5	*Enzyme lysate*	AVYLPNCDRKGFYKRKQCKPSR	-	Promote MAPK p38 and NF-κB	Increase the content of Ca^2+^ in mast cells, induce degranulation of mouse peritoneal mast cells	[161]
Murepavadin	Simulator	TWLKKRRWKKAKPP	IL-8 and CCL3 ↑	-	Induce mobilization and degranulation of LAD2 cells that express MrgprX2 endogenously and increase vascular permeability	[162]
AG-30/5C	Blood vessel	NH2-MLSLIFLHRLKSMRKRLDRKLRLWHRKNYP-COOH	LCT4, PGD2, PGE2, TNF-α, IL-8, MCP-1, MCP-3, MIP-1α, and MIP-1β ↑	Promote MAPK p38 and NF-κB	Activate mast cells degranulation and produce lipid mediators, enhance the chemotaxis of mast cells	[163]
Dermcidin	Sweat glands	-	CCL1, CCL2, IL-6, and TNF-α ↑	-	Activate mast cells	[164]
Pleurocidins	Fish	-	CCL2, CCL4, and MCP-1 ↑	-	-	[165]
Pleurocidins NCR-04	Fish	GWGSFFKKAAHVGKHVGKAALTHYL-NH2	PGD2 ↑	-	Promote LAD2 adhesion, migration, degranulation and release cysteine, LTs and PGD2	[165]

-: No description found. ↑: Increased levels. ↓: Reduced level.

**Table 5 pharmaceutics-15-02278-t005:** Outline of research on the regulatory effect of AMPs on DCs in recent years.

Peptide Name	Source	Amino Acid Sequence	Inflammatory Mediator	Signaling Pathway	Functions	Ref.
LL-37	Human	LLGDFFRKSKEKIGKEFKRIVQRIKDFLRNLVPRTES	IL-4, IL-6, IL-12, IL-10, and TNF-α ↑	-	Increase expression of HLA-DR and CD86 in immature DCs, enhance expansion and differentiation of DCs with CD103^+^/CD141^+^, and the antigen presenting ability of DCs, activate MDCs and promote differentiation	[168,169,170,171,172,173,174]
HBD2	Human	GIGDPVTCLKSGAICHPVFCPRRYKQIGTCGLPGTKCCKKP	INF-α and IFN-γ ↑	-	Activate plasmacytoid DCs by enhancing the uptake of CpG and self DNA	[175]
HBD3	Human	GIINTLQKYYCRVRGGRCAVLSCLPKEEQIGKCSTRGRKCCRRKK	INF-α ↑	-	Activate pDCs by enhancing the uptake of CpG and self DNA, phenotypic maturation of primary human skin migration DC induced by Langerhans cell like DC and human skin explants	[175,176]
MDF2β	Murine	CHTNGGYCVRAICPPSARRPGSCFPEKNPCCKYM	TNF-α and TNFR2 ↑	-	Induce co-stimulatory molecule upregulation and DCs maturation	[177,178]
MBD-14	Murine	FLPKTLRKFFCRIRGGRCAVLNCLGKEEQIGRCSNSGRKCCRKKK	-	-	Increase expression of CD40 and MHC Ⅱ on the surface of DCs, reduce the endocytosis ability	[179]
PMAP-23	Porcine	RIIDLLWRVRRPQKPKFVTVWVR	INF-α ↑	-	-	[180]
PMAP-36	Porcine	GRFRRLRKKTRKRLKKIGKVLKWIPPIVGSIPLGCG	INF-α ↑	-	-	[180]
Surfactin	*Bacilllus subtilis*	ELIVDIL	IL-6 and TNF-α ↑	Promote NF-κB	-	[181]
BEA	Fungi	-	IL-12 ↑	-	Active BMDCs	[182]
Pep19-2.5	Synthesis	GCKKYRRFRWKFKGKFWFWG	IL-6 ↓	-	Inhibit DCs migration	[183]

-: No description found. ↑: Increased levels. ↓: Reduced level.

**Table 6 pharmaceutics-15-02278-t006:** Outline of research on the regulatory effect of AMPs on neutrophils in recent years.

Peptide Name	Source	Amino Acid Sequence	Inflammatory Mediator	Signaling Pathway	Functions	Ref.
LL-37	Human	LLGDFFRKSKEKIGKEFKRIVQRIKDFLRNLVPRTES	TNF-α and IL-1β ↓ IL-8 and ROS ↑	Promote MAPK p38 and ERK	Inhibit neutrophil apoptosis, promote neutrophil to release NETs, and enhance the resistance of NETs to *Staphylococcus aureus* nuclease degradation	[185,186,187,189,190,191]
HBD-3	Human	GIINTLQKYYCRVRGGRCAVLSCLPKEEQIGKCSTRGRKCCRRKK	-	-	Inhibit neutrophils apoptosis	[192]
HBD-1	Human	DHYNCVSSGGQCLYSACPIFTKIQGTCYRGKAKCCK	-	-	Induce NET formation in PMNs	[193]
HBD-2	Human	GIGDPVTCLKSGAICHPVFCPRRYKQIGTCGLPGTKCCKKP	-	-	Limit the infiltration of neutrophils in the lungs	[194]
Scolopendrasin X	Centipede	MKKFHCLKKICKGLCAKL-CONH2	TNF-α and IL-6 ↓	-	Increase Ca^2+^ and superoxide anion in neutrophils, and migrate through G protein and phospholipase C pathway	[195]
Scolopendrasin IX	Centipede	MCKYFIKIVSKSAKK-CONH2	TNF-α, IL-6, IL-10, and CCL2 ↓	-	Increase Ca^2+^ and superoxide anion in neutrophils, and migrate through G protein and phospholipase C pathway, recruit neutrophils	[196]
Cathelicidin BF	*Bungarus fasciatus*	KFFRKLKKSVKKRAKEFFKKPRVIGVSIPF	CCL2, CXCL1, and CXCL2 ↑	-	Stimulate formation of NETs in neutrophils in vitro in a dose-dependent manner	[197,198,199]
LCN2	Neutrophils	-	IL-1α, IL-6, IL-8, and TNF-α ↑	Promote ERK1/2 and MAPK p38	-	[200]
KSLW	Synthesis	KKVVFWVKFK-NH2	-	-	Increase polymerization of F-Actin in neutrophils	[201]
Ac2-26	Synthesis	acetyl-AMVSEFLKQARFLENQEQEYVQAVK	CXCL2 ↑	-	-	[202]
Zd-14CFR	Synthesis	RGCRCNSKSFCVCR-NH2	TNF-α and IL-1β ↓	-	Decrease neutrophils infiltration	[203]

-: No description found. ↑: Increased levels. ↓: Reduced level.

**Table 7 pharmaceutics-15-02278-t007:** Outline of research on the regulatory effect of AMPs on eosinophils in recent years.

Peptide Name	Source	Amino Acid Sequence	Inflammatory Mediator	Signaling Pathway	Functions	Ref.
LL-37	Human	LLGDFFRKSKEKIGKEFKRIVQRIKDFLRNLVPRTES	IL-6, CXCL8, and CCL4 ↑	Promote ERK1/2	Trigger eosinophil nucleus to release Cys-LTs through FPRs, and release ECP	[205,206,207,208]

↑: Increased levels.

## Data Availability

Not applicable.

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
