# Peer review of "The Contribution of Antimicrobial Peptides to Immune Cell Function: A Review of Recent Advances"

_pharmaceutics, 2023, doi:10.3390/pharmaceutics15092278_

Round 1

Reviewer 1 Report

The manuscript entitled "The Contribution of Antimicrobial Peptides to Immune Cell 2 Function: A Review of Recent Advances" is a review of the evidence for the anti-inflammatory mechanisms of different antimicrobial peptides in immune cells, including macrophages, monocytes, lymphocytes, mast cells, dendritic cells, neutrophils, and eosinophils. The manuscript is clear and well-organized and could be used by researchers who study antimicrobial peptides.

There are a few points to clarify.

1 - Table 1, table 3, and table 6

There are some rows that don't have references. Please, clarify.

2 - Summary and Outlook

The current format of this section is something shallow. Please, improve it with some criticism and perspectives of the studies.

Author Response

Comments and Suggestions for Authors

The manuscript entitled "The Contribution of Antimicrobial Peptides to Immune Cell 2 Function: A Review of Recent Advances" is a review of the evidence for the anti-inflammatory mechanisms of different antimicrobial peptides in immune cells, including macrophages, monocytes, lymphocytes, mast cells, dendritic cells, neutrophils, and eosinophils. The manuscript is clear and well-organized and could be used by researchers who study antimicrobial peptides.

Response: Thank you very much for your evaluation of the paper. We appreciate the reviewer very much for their positive and constructive comments and suggestions. Those suggestions and comments are all valuable and very helpful for revising and improving our paper.

There are a few points to clarify.

1 - Table 1, table 3, and table 6

There are some rows that don't have references. Please, clarify.

Response: Thank you very much for your suggestions. The references have been supplemented in Table 1, Table 3, and Table 6, and in the corresponding position in the body of the revised manuscript.

2 - Summary and Outlook

The current format of this section is something shallow. Please, improve it with some criticism and perspectives of the studies.

Response: Thank you very much for your suggestions. In the original manuscript, we evaluated and criticized current research progress on AMPs' role in regulating immune cells as a whole, but were confused with recommendations for future research directions in the field. Based on your suggestions, we have made corresponding changes in the summary and outlook, please refer to the revised text in section Summary and Outlook.

Reviewer 2 Report

This review manuscript addresses the urgent need for novel antimicrobial agents due to the rise of multidrug-resistant microorganisms, focusing on the potential of antimicrobial peptides (AMPs) as alternatives to antibiotics. AMPs, found in various life forms, demonstrate strong antimicrobial activity against bacteria, fungi, parasites, and viruses, as well as promising anti-inflammatory, immunomodulatory, and anticancer properties. The review specifically highlights AMPs' role in regulating immune cells, such as macrophages, monocytes, lymphocytes, mast cells, dendritic cells, neutrophils, and eosinophils, discussing their abilities to activate immune cells, influence inflammatory mediators, and regulate inflammation-related signaling pathways. By shedding light on the diverse immunoregulatory characteristics of AMPs, the review contributes to a better understanding of their potential as therapeutic agents for combating microbial infections and controlling inflammatory conditions.

The manuscript is well-written and well-structured; however, there are some issues that need to be addressed before accepting it for publication:

Section 2.4, "Regulation of macrophages by plant-derived AMPs," is too brief and needs to be expanded to provide more comprehensive insights.

The review lacks a deeper understanding of the specificity of AMPs. In the Introduction, it is mentioned that AMPs interact with negatively charged phospholipids and nucleic acids, but it does not explain why non-mammalian derived AMPs have low cytotoxicity for human cells which also contain negatively charged phospholipids and nucleic acids. The authors should address how and why evolutionarily non-animal AMPs interact with receptors of mammalian cells, leading to beneficial effects for the host and inhibitory effects on pathogens. 

Author Response

Comments and Suggestions for Authors

This review manuscript addresses the urgent need for novel antimicrobial agents due to the rise of multidrug-resistant microorganisms, focusing on the potential of antimicrobial peptides (AMPs) as alternatives to antibiotics. AMPs, found in various life forms, demonstrate strong antimicrobial activity against bacteria, fungi, parasites, and viruses, as well as promising anti-inflammatory, immunomodulatory, and anticancer properties. The review specifically highlights AMPs' role in regulating immune cells, such as macrophages, monocytes, lymphocytes, mast cells, dendritic cells, neutrophils, and eosinophils, discussing their abilities to activate immune cells, influence inflammatory mediators, and regulate inflammation-related signaling pathways. By shedding light on the diverse immunoregulatory characteristics of AMPs, the review contributes to a better understanding of their potential as therapeutic agents for combating microbial infections and controlling inflammatory conditions.

Response: Thank you very much for your evaluation of the paper. We appreciate the reviewer very much for their positive and constructive comments and suggestions. Those suggestions and comments are all valuable and very helpful for revising and improving our paper.

The manuscript is well-written and well-structured; however, there are some issues that need to be addressed before accepting it for publication:

Section 2.4, "Regulation of macrophages by plant-derived AMPs," is too brief and needs to be expanded to provide more comprehensive insights.

Response: Thank you very much for your suggestions. There are 305 of plant-derived AMPs that have been discovered (https://dbaasp.org/search). We searched again for the literature on the role of plant-derived AMPs in immune cell function (macrophages, monocytes, lymphocytes, mast cells, dendritic cells, neutrophils, and eosinophils). Only PN5 has been reported to study the effects of AMPs on immune function of macrophages.

The review lacks a deeper understanding of the specificity of AMPs. In the Introduction, it is mentioned that AMPs interact with negatively charged phospholipids and nucleic acids, but it does not explain why non-mammalian derived AMPs have low cytotoxicity for human cells which also contain negatively charged phospholipids and nucleic acids. The authors should address how and why evolutionarily non-animal AMPs interact with receptors of mammalian cells, leading to beneficial effects for the host and inhibitory effects on pathogens.

Response: Thank you very much for your suggestions. The central question in the research of AMPs is how AMPs these peptides specifically target the invading pathogen while spare the host cells? The differences in the composition of cell membrane between the pathogens and the host cells have been considered to underpin the targeting specificity of AMPs ((Dawson RM et al., Crit Rev Microbiol, 2008; Sohlenkamp C et al., FEMS Microbiol Rev, 2016; Zhang QY et al., Mil Med Res, 2021). Outer membrane of prokaryotic cell is negatively charged owing to presence of lipopolysaccharides or teichoic acid, whereas the outer leaflet of eukaryotic cell consists of zwitterionic phosphatidylcholine and sphingomyelin phospholipids. Mammalian cell membranes contain cholesterol, which can enhance cell stability, prevent and treat the insertion of antimicrobial peptides, and destroy cells (Erdem Büyükkiraz M, J Appl Microbiol, 2022). But in actual fact some natural AMPs have a certain degree of hemolysis of red blood cells. Fortunately, a structural optimization strategy with chemical synthesis was implemented to overcome the potential toxicity.

Reviewer 3 Report

1.       What are the future implication of this review to the science? Please justify, discuss, and include the summarized line at the end of the abstract.

2.       This review needs updates as it is missing the bacterial origin AMPs (bacteriocins or lantibiotics) immunomodulatory properties. Just Google the words and there is a lot of information about the same. The authors need to update this review.

3.       Are there any AMPs that are under clinical trial regarding the use of their immunomodulatory properties? It will be nice if the authors can summarize a table about the current clinical trial of AMPs in a similar scenario.

4.       There are many limitations associated with AMPs in the further translation to clinical settings. The authors need to summarize the limitations of the current review in a separate section.

Minor edits are required. 

Author Response

Comments and Suggestions for Authors

  1. What are the future implication of this review to the science? Please justify, discuss, and include the summarized line at the end of the abstract.

Response: Thank you very much for your suggestions. We've described the scientific and practical implications of this review, please check out the revised text in the end of the abstract and Summary and Outlook.

  1. This review needs updates as it is missing the bacterial origin AMPs (bacteriocins or lantibiotics) immunomodulatory properties. Just Google the words and there is a lot of information about the same. The authors need to update this review.

Response: Thank you very much for your suggestions. There are 540 of bacteria-derived AMPs that have been discovered (https://dbaasp.org/search). Regarding the issue you mentioned, we searched again for the literature on the role of bacteria-derived AMPs in immune cell function (macrophages, monocytes, lymphocytes, mast cells, dendritic cells, neutrophils, and eosinophils). About 10 AMPs-derived from bacteria has been reported to study the effects of AMPs on immune function of macrophages. We've added new relevant information, please check out the section '2.5. Regulation of macrophages by microbial-derived AMPs' and section '6.2. Regulation of dendritic cells by microbial-derived AMPs'. In the meantime, 2 of virus-derived AMPs, 170 of fungi-derived AMPs, and 2 of protisa-derived AMPs were used to search to prevent missing information.

  1. Are there any AMPs that are under clinical trial regarding the use of their immunomodulatory properties? It will be nice if the authors can summarize a table about the current clinical trial of AMPs in a similar scenario.

Response: Thank you very much for your suggestions. We added a separate section '9. Application prospects of AMPs' where we summarized the current application of AMPs. According to Data Repository of Antimicrobial Peptides (DRAMP) Database (http://dramp.cpu-bioinfor.org/, last updated on 2023-7-4), 96 AMPs have been devel-oped as drug candidates for the peptide drug market, as shown in Supplementary Table 1. CZEN-002(Melantropin, (CKPV)2), EA-230, RDP58(Delmitide), Ghrelin, and EA360 are under clinical trial regarding the use of their immunomodulatory properties.

  1. There are many limitations associated with AMPs in the further translation to clinical settings. The authors need to summarize the limitations of the current review in a separate section.

Response: Thank you very much for your suggestions. We added a separate section '9. Application prospects of AMPs' where we summarized the current application of AMPs and some limitations associated with AMPs in the further translation to clinical settings, please refer to the supplemented article.

Comments on the Quality of English Language

Minor edits are required.

Response: Thank you very much for your suggestion. The original manuscript was edited by American Journal Experts on July 12, 2023 (the verification code: 4819-786D-C335-3E46-01DP), and the revised manuscript was also re-edited by American Journal Experts on August 26, 2023.

We appreciate the reviewer very much for their positive and constructive comments and suggestions. Those suggestions and comments are all valuable and very helpful for revising and improving our paper.

Round 2

Reviewer 3 Report

The authors successfully responded to the reviewer's comments and updated the manuscript as well. 

This is fine.